# Rethinking Federated Prompt Learning for Medical Images: From Textual Tuning to Visual Manifold Anchoring

**Yipan Wei**[1]  **Wenke Huang**[2][†]  **Yapeng Li**[1][†]  **He Li**[1]  **Qixin Zhang**[1]  **Mang Ye**[1]  **Bo Du**[1][†]

## Abstract

Federated Prompt Learning (FPL) adapts Vision-Language Models to privacy-sensitive medical imaging, typically via a "textual tuning" paradigm that assumes the frozen visual encoder provides a discriminative feature geometry. We argue this assumption breaks down in medical settings, leading to two geometric pathologies: (1) **Intra-client: Medical Manifold Collapse**, where high morphological similarity reduces the *effective rank* of visual features; and (2) **Inter-client: Medical Topological Misalignment**, where heterogeneous acquisition protocols induce inconsistent *geometry* across clients. To address these, we propose **FedMAP**, which shifts the paradigm to Visual Manifold Anchoring. FedMAP utilizes an LLM-derived codebook as a client-invariant synchronization signal to restructure the visual space, via **Manifold Semantic Anchoring (MSA)** and **Topology Structural Alignment (TSA)** to enforce consistent inter-class relations. Experiments on FedISIC, FedCamelyon17, and a private ultrasound dataset show that FedMAP consistently outperforms state-of-the-art methods, especially under cross-center heterogeneity, where frozen visual geometry is strongly distorted. Code is available at https://github.com/YipanWei/FedMAP.

[†]Co-corresponding authors. [1]School of Computer Science, National Engineering Research Center for Multimedia Software, Institute of Artificial Intelligence, Hubei Key Laboratory of Multimedia and Network Communication Engineering, Wuhan University, Wuhan, China [2]College of Computing and Data Science, Nanyang Technological University, Singapore. Correspondence to: Wenke Huang <wenke.huang@ntu.edu.sg>, Yapeng Li <yapengli@whu.edu.cn>, Bo Du <dubo@whu.edu.cn>.

*Proceedings of the 43rd International Conference on Machine Learning*, Seoul, South Korea. PMLR 306, 2026. Copyright 2026 by the author(s).

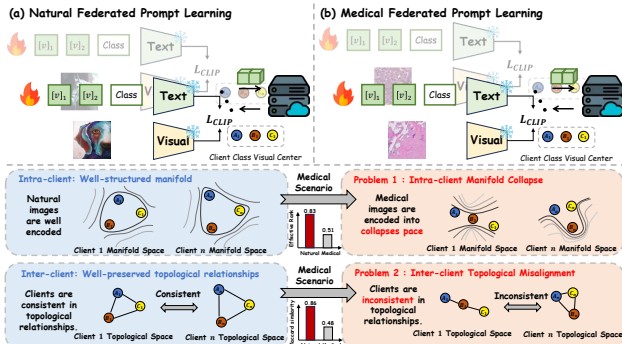

*Figure 1.* **Motivation. Left: Natural federated prompt learning.** Clients form well-structured class manifolds and share aligned neighborhood relations, so text-side tuning can separate classes effectively. **Right: Medical federated prompt learning.** Medical images often compress class features into narrow subspaces and create inconsistent neighborhood structures across centers, motivating visual manifold anchoring.

## 1. Introduction

**Federated Prompt Learning (FPL)** has emerged as a practical paradigm for adapting large foundation models (Radford et al., 2021; Li et al., 2023a; Wu et al., 2024; Zhang et al., 2024b; Nath et al., 2025) to downstream tasks in distributed and privacy-sensitive settings (Guo et al., 2023b; Lu et al., 2023). Building on Federated Learning (FL) (McMahan et al., 2017; Shi & Lipani, 2024; Li et al., 2020a; Huang et al., 2024), FPL keeps the pretrained backbone frozen and updates only lightweight prompt parameters (Liao et al., 2025; Fang et al., 2025), resulting in low communication overhead through compact prompt exchange (Jia et al., 2022; Zhao et al., 2023; Singha et al., 2025; Yang et al., 2025). This design is particularly attractive for **medical image analysis**, where patient data must remain local and computational resources are often constrained (Shokri & Shmatikov, 2015; Yang et al., 2019), while pretrained knowledge can help alleviate local data scarcity (Sheller et al., 2020; Kaissis et al., 2020; Wang et al., 2022b; Zhou et al., 2022a).

Despite its success on natural images, transferring FPL to medical data remains challenging (Moor et al., 2023; Khattak et al., 2023). A key reason is the prevailing **"textual tuning"** paradigm (Guo et al., 2023b; Fang et al., 2025; Li et al., 2024a; Yang et al., 2025), which freezes the visual encoder and adapts primarily by shifting *text-side* decision

boundaries, implicitly assuming that the frozen visual features already exhibit a well-structured geometry (Papyan et al., 2020; Galanti et al., 2022; Thrampoulidis et al., 2022). In medical federated settings, however, subtle morphological differences and cross-center acquisition shifts can substantially distort this geometry (Li et al., 2023c; Jing et al., 2021). Consequently, purely shifting textual boundaries can be unreliable: text-induced classifiers may fail when *intra-class* discriminative directions collapse, and global aggregation can become unstable when *inter-class* relations differ across clients (Raghu et al., 2019; Du et al., 2025; Shi et al., 2023).

As illustrated in **Figure 1**, we summarize these challenges as two geometric pathologies of the frozen visual backbone:

- **(1) Intra-client: Medical Manifold Collapse.** High inter-class morphological similarity can compress each class manifold into a narrow, low-dimensional region (Wu et al., 2025), as shown in the right panel of Figure 1, leading to a **reduced intra-class effective rank** and limited discriminative degrees of freedom for text-side separation.

- **(2) Inter-client: Medical Topological Misalignment.** Heterogeneous devices and protocols perturb neighborhood relations (Li et al., 2021b), as shown in the right panel of Figure 1; without a shared reference, clients may converge to **conflicting local optima**, yielding inconsistent **inter-class geometry** that hinders global consensus (Karimireddy et al., 2020; Qu et al., 2022).

To address these issues, we propose **FedMAP** (**Fed**erated **M**anifold-**A**nchored **P**rompting), which shifts FPL **from textual tuning to visual manifold anchoring** (Jia et al., 2022). The key idea is to introduce a *client-invariant* structural reference that remains available even when global visual data are inaccessible. Specifically, we employ a large language model to expand class labels into diverse attributes and external class-name semantics (Menon & Vondrick, 2023; Pratt et al., 2023; Tian et al., 2024; Li et al., 2024d) and construct a knowledge-enriched semantic codebook, which serves as a **static synchronization signal** for the shared class topology (Li et al., 2025). FedMAP then injects learnable prompts into the visual encoder and regularizes the geometry via two mechanisms: **Manifold Semantic Anchoring (MSA)** counteracts intra-client rank deficiency by anchoring visual features to the codebook, thereby expanding the effective subspace, and **Topology Structural Alignment (TSA)** mitigates inter-client misalignment by aligning local visual relations with the global linguistic structure.

Our main contributions are summarized as follows:

❶ *Geometric Diagnosis.* We identify two failure modes in medical federated prompt learning, intra-client medi-

cal manifold collapse and inter-client medical topological misalignment, clarifying why text-only tuning breaks when visual features collapse and client relations drift.

❷ *Visual Manifold Anchoring.* We propose **FedMAP**, which introduces an attribute codebook expanded by a large language model as a client-invariant semantic reference and shifts federated prompt learning toward visual manifold anchoring with Manifold Semantic Anchoring and Topology Structural Alignment.

❸ *Empirical Validation.* Experiments on FedISIC (Ogier du Terrail et al., 2022a), FedCamelyon17 (Koh et al., 2021), and a private dataset show that FedMAP consistently outperforms prior methods under cross-center heterogeneity.

## 2. Related Work

### 2.1. Federated Learning

Federated Learning (FL) enables collaborative training across distributed clients without sharing raw data. FedAvg (McMahan et al., 2017) averages client updates to learn a global model, but its performance can degrade under non-IID data. Prior work addresses statistical heterogeneity via: (1) *regularizing local updates:* FedDyn (Acar et al., 2021), FedACG (Kim et al., 2024), FedNTD (Lee et al., 2022), and FedRS (Li & Zhan, 2021); (2) *improving aggregation:* SCAFFOLD (Karimireddy et al., 2020), FedNova (Wang et al., 2020), and FedOPT (Reddi et al., 2021); (3) *aligning representations:* MOON (Li et al., 2021a), FedProto (Tan et al., 2022), FCCL/FCCL+/FPL (Huang et al., 2022; 2023a;b), FedTA (Yu et al., 2025), and FedFA (Zhou et al., 2023). In real deployments, domain shift remains a major bottleneck, motivating methods such as FedBN (Li et al., 2021b), FDSE (Wang et al., 2025), HarmoFL (Jiang et al., 2022), FedGA (Zhang et al., 2023), and FedDG (Liu et al., 2021), generalized FL mechanisms such as FedMut and dynamic parameter reset (Hu et al., 2024; Wu et al., 2026), as well as personalization strategies (Sun et al., 2021).

However, traditional FL typically exchanges full model parameters, incurring **substantial communication costs** for foundation models in resource-constrained medical environments. FedMAP instead leverages FPL to reduce communication overhead while improving robustness.

### 2.2. Federated Prompt Learning

Prompt learning has emerged as a parameter-efficient fine-tuning (PEFT) technique for adapting large vision-language models (VLMs), alongside broader deep visual advances in structured imaging representation learning (Li et al., 2024e). Early approaches relied on manual templates (CLIP (Radford et al., 2021)), which later evolved into learnable con-

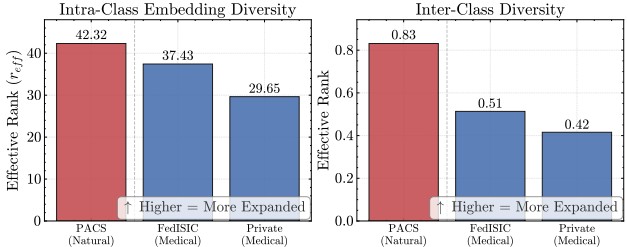

*Figure 2.* **Statistical Evidence of Intra-client Manifold Collapse.** We analyze the geometry of frozen CLIP features across Natural (PACS) and Medical domains. Medical datasets exhibit lower **(Left) Intra-class Effective Rank** and **(Right) normalized Inter-class Effective Rank**.

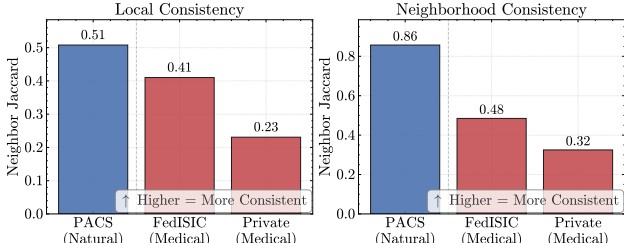

*Figure 3.* **Statistical Evidence of Inter-client Topological Misalignment.** We quantify the structural consistency of feature spaces across clients using Local and Neighborhood Consistency metrics, based on Jaccard similarity of nearest neighbors.

tinuous prompts such as CoOp (Zhou et al., 2022b), Co-CoOP (Zhou et al., 2022a), and KgCoOp (Yao et al., 2023).

FPL integrates PEFT into FL to alleviate the communication bottleneck by exchanging only lightweight prompt parameters: clients update prompts locally, and the server aggregates these prompt updates. **Text-centric methods** such as FedPrompt (Zhao et al., 2023) and PromptFL (Guo et al., 2023b) follow FedAvg to learn text prompts, offering a simple yet efficient solution. To better cope with client heterogeneity, personalized and model-heterogeneous variants such as pFedPrompt (Guo et al., 2023a), pFedPG (Yang et al., 2023), FedAPT (Su et al., 2024), pFedLoRA (Yi et al., 2026b), and pFedMoE (Yi et al., 2026a) maintain client-specific components, while decoupled approaches such as FedOTP (Li et al., 2024a), FedPGP (Cui et al., 2024), and FedPHA (Fang et al., 2025) partition prompts into role-specific modules. This design improves adaptability under non-IID and cross-site shifts without incurring the cost of full-model exchange. Recent work also improves robustness under OOD or view perturbations, such as FO-CoOP (Liao et al., 2025) and robust multi-view learning (Li et al., 2024c), or leverages richer multimodal context, such as FedMVP (Singha et al., 2025) and audio-visual zero-shot learning (Li et al., 2023b).

Most FPL methods rely on "textual tuning" or passive visual adapters and thus lack a global structural blueprint. These approaches may fail to reconstruct discriminative topology under severe **Intra-client Manifold Collapse** in medical data. In contrast, FedMAP adopts active **Visual Manifold Anchoring** and uses a global semantic codebook to explicitly expand the collapsed visual manifold.

### 2.3. Medical Federated Learning

Medical Federated Learning (MedFL) is challenged by strong cross-institution heterogeneity (Rieke et al., 2020; Adnan et al., 2022). Prior work studies (i) representation alignment, concept-shift handling, and fairness-aware aggregation (Jiang et al., 2022; Yan et al., 2020; 2024; Jiang et al., 2023; Wei et al., 2025), (ii) data-efficient training under limited and heterogeneous supervision, such as federated active

learning, label-efficient SSL, mixed supervision, and dataset distillation (Chen et al., 2024; Yan et al., 2023; Wicaksana et al., 2022; Jin et al., 2025), (iii) multimodal/EHR collaboration without sharing raw records (Brisimi et al., 2018; Vaid et al., 2021; Wang et al., 2022a), and (iv) leveraging textual or clinical semantic priors and medical VLM knowledge transfer for robustness (Poudel et al., 2024; Zhu et al., 2024; Li et al., 2024b; Huang et al., 2026; Wei et al., 2026). Privacy/security and unlearning are also increasingly important (Salim & Park, 2022; Deng et al., 2024; Sahoo et al., 2024). The community further pushes standardized evaluation and scalable platforms via realistic suites/benchmarks and system-level settings (Ogier du Terrail et al., 2022b; Alhamoud et al., 2024; Zhang et al., 2024a; Wang et al., 2024; Xie et al., 2025; 2024).

While recent methods leverage language priors for augmentation, they rarely utilize LLMs as a **structural blueprint** for the feature space. FedMAP employs an LLM-derived codebook as a **client-invariant synchronization signal**, enforcing topological alignment to resolve inter-client geometric mismatches.

## 3. Methodology

In this section, we first formally characterize the geometric pathologies—**Manifold Collapse** and **Topological Misalignment**—inherent in the frozen VLM feature space when applied to medical data. To rectify these intrinsic distortions, we propose **FedMAP**, a two-stage framework consisting of **MSA** and **TSA**, as illustrated in Figure 4.

### 3.1. Preliminaries

**Problem Formulation.** We consider a standard FL system comprising $K$ decentralized clients, indexed by $k \in \{1, \dots, K\}$. Each client holds a private local dataset $\mathcal{D}_k = \{(x_i, y_i)\}_{i=1}^{N_k}$, where $x_i \in \mathcal{X}$ is the input medical image and $y_i \in \mathcal{Y} = \{1, \dots, C\}$ is the corresponding label. In FedMAP, we keep the backbone weights $\Theta_{frozen}$ fixed and collaboratively optimize only lightweight parameters: a set of **visual prompts P**.

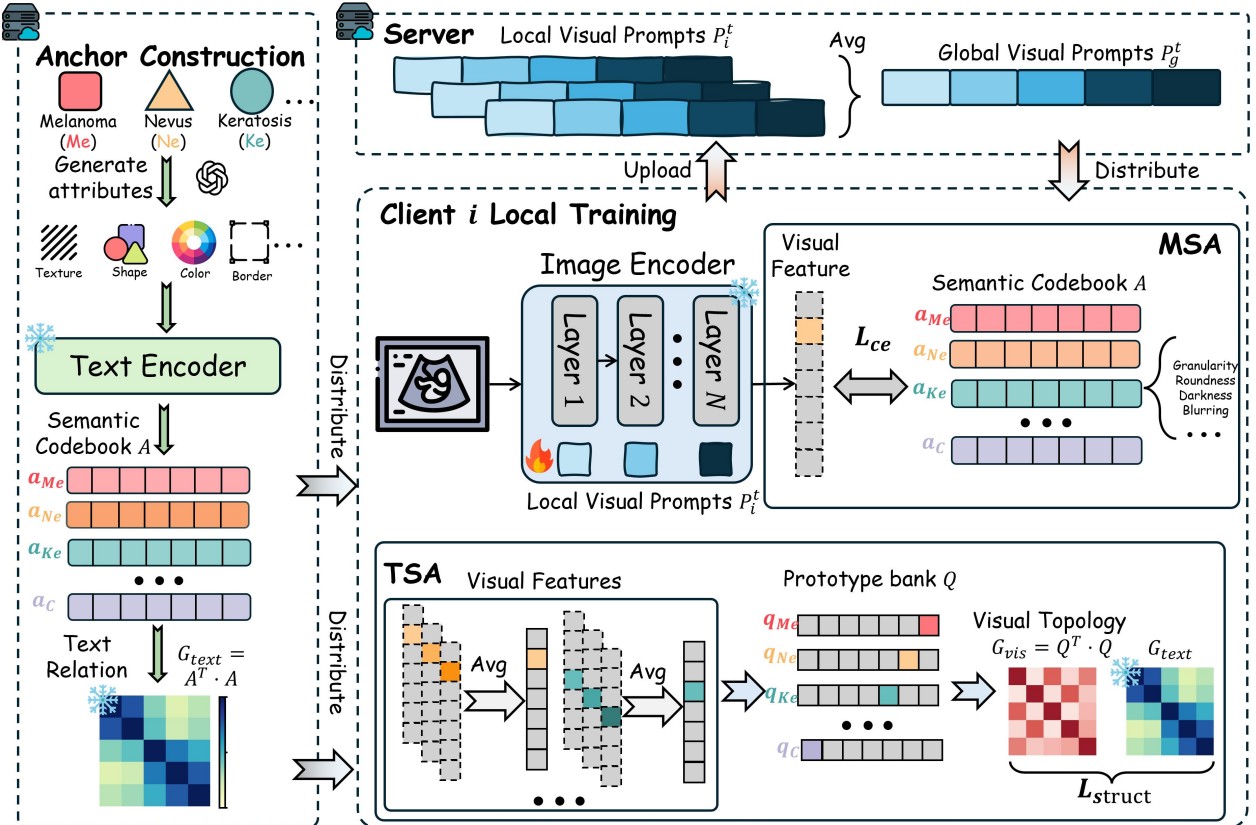

**Figure 4.** Overview of FedMAP. The server first constructs an LLM-derived semantic codebook $A$ and text relation matrix $G_{\text{text}}$ from class labels. During federated training, each client injects one depth-aware visual prompt per image-encoder layer and optimizes MSA and TSA losses to anchor visual features and align class topology. Only visual prompts are uploaded, averaged on the server, and redistributed.

**Deep Visual Prompting.** We adopt a VLM backbone consisting of an image encoder $\mathcal{E}_I$ and a text encoder $\mathcal{E}_T$. We maintain multi-layer visual prompts $\mathbf{P} = \{\mathbf{P}^{(\ell)}\}_{\ell \in \mathcal{L}_p}$, where $\mathbf{P}^{(\ell)} \in \mathbb{R}^{m_p \times D}$ denotes $m_p$ learnable visual prompt tokens inserted at transformer block $\ell$, and $\mathcal{L}_p$ specifies the prompted layers. At each prompted block, we prepend $\mathbf{P}^{(\ell)}$ to the patch-token sequence; we denote this token augmentation operator as $[x, \mathbf{P}]$.

For an input $x$, the adapted visual representation is:

$$\mathbf{v} = \mathcal{E}_I([x, \mathbf{P}]; \Theta_{frozen}), \quad (1)$$

where $\Theta_{frozen}$ remains fixed throughout training. In federated round $t$, we denote the server prompt by $\mathbf{P}_g^t$ and the local prompt by $\mathbf{P}_k^t$.

The overarching objective is to minimize the global empirical risk without exposing $\mathcal{D}_k$:

$$\min_{\mathbf{P}} \mathcal{L}_{global}(\mathbf{P}) = \sum_{k=1}^{K} \frac{N_k}{N} \mathbb{E}_{(x,y) \sim \mathcal{D}_k}[\mathcal{L}_{CE}(f(x, \mathbf{P}; \tau), y)],$$

$$(2)$$

where $N = \sum_{k=1}^{K} N_k$ is the total number of samples, and $f(\cdot)$ is the prediction head.

### 3.2. Motivation

While FPL is parameter-efficient, we find that *feature geometry* degenerates severely in medical FL. We characterize two key pathologies with quantitative statistics.

**(1) Intra-client: Medical Manifold Collapse.** Unlike natural images with semantically separable classes, medical categories often differ by subtle morphology and texture. As shown in Figure 2, we diagnose intra-client collapse using two complementary statistics: **Intra-class Effective Rank** and **normalized Inter-class Effective Rank**. Specifically, frozen CLIP features on medical datasets exhibit lower intra-class effective rank than natural images, decreasing from 42.32 on PACS to 37.43 on FedISIC and 29.65 on Private. Meanwhile, the normalized inter-class effective rank drops from 0.83 to 0.51/0.42, suggesting reduced diversity of class-wise prototypes/centers and a more entangled class geometry. This collapse compresses class manifolds into narrow subspaces ($r_{\text{eff}} \ll D$), leaving limited geometric degrees of freedom and making standard text-only tuning poorly conditioned for fine-grained discrimination.

**(2) Inter-client: Medical Topological Misalignment.** Due to scanner/protocol heterogeneity, different clients induce different local neighborhood structures in the embedding

space. We quantify cross-client structural inconsistency in Figure 3 via **Jaccard Similarity**-based neighborhood consistency, where medical datasets show substantially lower stability than natural images. As a result, clients optimize toward *incompatible* geometries, and aggregating their updates can cause destructive interference, hindering a coherent global decision boundary.

### 3.3. Manifold Semantic Anchoring (MSA)

To mitigate the **Intra-Client: Medical Manifold Collapse** caused by limited and biased local data, MSA introduces a set of *semantic anchors* as a frozen, knowledge-enriched reference in the joint CLIP embedding space. Rather than using a single potentially ambiguous class name, we construct *attribute-augmented anchors* that provide a more stable and discriminative supervision signal for medical categories.

#### 3.3.1. HIERARCHICAL VISUAL ATTRIBUTE DISTILLATION

Instead of relying on generic class names that may lack visually discriminative cues, we leverage LLMs to produce fine-grained visual attributes and construct a semantic codebook. To improve visual relevance and reduce linguistic noise, we adopt a **coarse-to-fine elicitation** strategy.

For each category $c \in \mathcal{Y}$, the attribute generation consists of:

1. **Visual Interrogation:** query the LLM to describe salient visual appearance of category $c$, producing a diverse attribute pool;

2. **Structured Summarization:** distill the pool into $M$ concise and non-overlapping attributes $\mathcal{S}_c = \{s_{c,1}, s_{c,2}, \ldots, s_{c,M}\}$.

This hierarchical process yields attributes that are semantically rich while remaining closely tied to lesion appearance.

#### 3.3.2. ROBUST ANCHOR CONSTRUCTION VIA SEMANTIC ENSEMBLE

We encode each attribute prompt into the shared CLIP space with the frozen text encoder $\mathcal{E}_T$. For each $s_{c,k} \in \mathcal{S}_c$, we use a structured template $\phi(c, s_{c,k})$ formatted as "$\{c\} : \{s_{c,k}\}$", which encourages the encoder to focus on class-attribute association. We then compute an ensemble anchor by averaging and $\ell_2$-normalizing:

$$\mathbf{a}_c^{raw} = \frac{1}{M} \sum_{m=1}^{M} \mathcal{E}_T(\phi(c, s_{c,m})), \qquad \mathbf{a}_c = \frac{\mathbf{a}_c^{raw}}{\|\mathbf{a}_c^{raw}\|_2}. \tag{3}$$

We stack all anchors as $\mathbf{A} = [\mathbf{a}_1, \ldots, \mathbf{a}_C] \in \mathbb{R}^{d \times C}$. Crucially, $\mathbf{A}$ is **frozen** during federated training and serves as a semantic prior.

Given a visual feature $\mathbf{v}$, we compute the anchor-based predictive distribution:

$$p(y = c|x) = \frac{\exp(\mathrm{sim}(\mathbf{v}, \mathbf{a}_c)/\tau)}{\sum_{j=1}^{C} \exp(\mathrm{sim}(\mathbf{v}, \mathbf{a}_j)/\tau)}, \tag{4}$$

where $\mathrm{sim}(\cdot, \cdot)$ denotes cosine similarity and $\tau$ is the temperature. The supervised objective $\mathcal{L}_{CE}$ is computed using the above distribution, thereby anchoring visual representations to a fixed semantic reference.

### 3.4. Topology Structural Alignment (TSA)

While MSA provides an absolute semantic reference via frozen anchors, it does not explicitly constrain the *pairwise similarity structure* among categories. Due to domain shift and statistical heterogeneity, different clients may learn inconsistent inter-class geometry. TSA addresses this issue by aligning the *class-wise similarity structure* in the visual space with a *text-derived relation matrix*.

#### 3.4.1. TEXT RELATION MATRIX

We define a static **text relation matrix** from semantic anchors:

$$\mathbf{G}_{text}[i, j] = \mathbf{a}_i^\top \mathbf{a}_j, \quad \forall i, j \in \{1, \ldots, C\}. \tag{5}$$

Since anchors are attribute-augmented and frozen, $\mathbf{G}_{text}$ provides a client-invariant reference of semantic affinity among categories.

#### 3.4.2. VISUAL PROTOTYPE BANK AND VISUAL RELATION MATRIX

Each client maintains a prototype bank $\mathbf{Q} = \{\mathbf{q}_1, \ldots, \mathbf{q}_C\}$ on the unit hypersphere, updated by EMA using observed samples. Let $\bar{\mathbf{v}}_c$ be the mean feature of class $c$ in a mini-batch; we update:

$$\mathbf{q}_c \leftarrow \frac{\mu \mathbf{q}_c + (1 - \mu) \, \mathrm{sg}(\bar{\mathbf{v}}_c)}{\|\mu \mathbf{q}_c + (1 - \mu) \, \mathrm{sg}(\bar{\mathbf{v}}_c)\|_2}, \tag{6}$$

where $\mu$ is the momentum and $\mathrm{sg}(\cdot)$ denotes stop-gradient.

We then form the **visual relation matrix** as the Gram matrix of prototypes:

$$\mathbf{G}_{vis}[i, j] = \mathbf{q}_i^\top \mathbf{q}_j, \quad \mathbf{G}_{vis} \in \mathbb{R}^{C \times C}. \tag{7}$$

#### 3.4.3. RELATIONAL STRUCTURE DISTILLATION

TSA aligns the visual class-wise similarity structure to the text-derived reference by minimizing:

$$\mathcal{L}_{struct} = \|\mathbf{G}_{vis} - \mathbf{G}_{text}\|_F^2. \tag{8}$$

In practice, to reduce computation and avoid unobserved classes, we apply the loss on the subset of classes that appear on the client, i.e., on the corresponding rows/columns of $\mathbf{G}_{vis}$ and $\mathbf{G}_{text}$.

*Table 1.* Comparative results on **FedISIC**. We report Accuracy/Macro-F1 (%) over three seeds for trainable methods; Zero-Shot CLIP is deterministic.

| Methods | MSK | | VI-MX | | VI-DL | | RS | | VI-HD | | BCN | | Avg | |
|---|---|---|---|---|---|---|---|---|---|---|---|---|---|---|
| | Acc | F1 | Acc | F1 | Acc | F1 | Acc | F1 | Acc | F1 | Acc | F1 | Acc | F1 |
| Zero-Shot CLIP [ICML 2021] | 16.51 | 4.70 | 0.38 | 0.70 | 6.10 | 4.00 | 11.73 | 3.30 | 0.00 | 0.00 | 0.00 | 0.00 | 5.79 | 2.12 |
| PromptFL [TMC 2023] | 58.75±0.72 | 35.23±1.17 | 94.10±0.27 | 41.90±1.23 | 67.17±0.21 | 42.90±0.44 | 50.96±0.25 | 28.17±0.40 | 46.94±2.12 | 40.13±2.85 | 72.72±5.20 | 51.07±18.97 | 65.11±0.48 | 39.90±2.82 |
| PromptFL+Prox [PMLR 2020] | 57.94±0.59 | 33.67±0.97 | 93.72±0.32 | 40.93±4.32 | 67.51±0.31 | 43.40±1.04 | 50.29±0.13 | 27.43±0.58 | 47.36±1.76 | 39.43±2.55 | 77.65±1.31 | 52.07±9.98 | 65.75±0.47 | 39.49±1.02 |
| FedKgCoOP [CVPR 2023] | 53.15±0.16 | 25.20±0.20 | 93.26±0.19 | 25.57±2.40 | 62.00±1.38 | 28.10±3.47 | 44.62±1.05 | 21.30±1.82 | 42.89±0.35 | 33.13±1.18 | 80.68±3.01 | 33.07±1.20 | 62.77±0.14 | 27.73±1.30 |
| FedCLIP [ICLR 2023] | 58.56±0.28 | 32.33±0.76 | 93.47±0.19 | 27.33±1.67 | 66.37±1.03 | 37.07±1.97 | 52.06±0.84 | 27.77±0.85 | 44.51±2.11 | 37.73±5.05 | 81.82±1.14 | 38.03±0.59 | 66.13±0.38 | 33.38±1.28 |
| FedAPT [AAAI 2024] | 58.71±1.70 | 37.20±2.35 | 94.06±0.33 | 39.30±2.59 | 67.51±0.31 | 43.00±0.66 | 51.40±1.14 | 28.20±1.14 | 45.33±3.07 | 35.70±4.90 | 72.73±6.01 | 56.07±3.88 | 64.95±0.40 | 39.91±0.58 |
| FOCoOP [ICML 2025] | 44.35±5.30 | 17.53±5.01 | 72.90±19.48 | 23.55±2.47 | 53.67±4.49 | 19.90±7.01 | 35.84±3.34 | 15.33±4.60 | 41.06±3.36 | 32.67±9.13 | 61.36±11.86 | 24.73±2.03 | 51.53±6.50 | 22.13±4.63 |
| FedMVP [ICCV 2025] | 52.96±3.35 | 24.90±2.10 | 93.38±0.98 | 23.47±5.04 | 63.54±1.42 | 32.33±9.69 | 43.95±2.92 | 19.70±2.41 | 39.23±1.76 | 22.50±0.44 | 78.79±2.37 | 33.33±1.12 | 61.98±0.51 | 26.04±2.26 |
| **FedMAP (Ours)** | **71.04±3.28** | **58.53±5.09** | **97.60±0.43** | **72.10±3.00** | **79.52±2.20** | **68.17±4.12** | **60.31±0.69** | **44.20±4.52** | **50.20±5.53** | **38.57±6.01** | **81.07±2.86** | **75.23±2.84** | **73.29±1.29** | **59.47±1.98** |

*Table 2.* Comparative results on **FedCamelyon17**. We report Accuracy/Macro-F1 (%) over three seeds for trainable methods; Zero-Shot CLIP is deterministic.

| Methods | CWZ | | RST | | UMCU | | RUMC | | LPON | | Avg | |
|---|---|---|---|---|---|---|---|---|---|---|---|---|
| | Acc | F1 | Acc | F1 | Acc | F1 | Acc | F1 | Acc | F1 | Acc | F1 |
| Zero-Shot CLIP [ICML 2021] | 59.52 | 53.30 | 51.83 | 42.20 | 68.78 | 66.00 | 55.29 | 47.40 | 51.75 | 37.20 | 57.43 | 49.22 |
| PromptFL [TMC 2023] | 91.13±0.03 | 91.10±0.00 | 85.05±0.19 | 84.97±0.23 | 90.93±0.29 | 90.90±0.26 | 90.23±0.75 | 90.20±0.79 | 93.96±1.15 | 93.93±1.17 | 90.26±0.35 | 90.22±0.35 |
| PromptFL+Prox [PMLR 2020] | 91.11±0.07 | 91.07±0.06 | 85.05±0.18 | 85.00±0.17 | 90.90±0.28 | 90.87±0.25 | 90.40±0.61 | 90.37±0.64 | 94.01±1.20 | 94.00±1.25 | 90.29±0.36 | 90.26±0.36 |
| FedKgCoOP [CVPR 2023] | 90.71±0.06 | 90.70±0.10 | 74.53±2.71 | 73.53±3.26 | 89.51±0.09 | 89.50±0.10 | 86.27±0.83 | 86.10±0.89 | 92.42±0.99 | 92.43±1.01 | 88.70±0.32 | 88.67±0.33 |
| FedCLIP [ICLR 2023] | 90.13±0.28 | 90.10±0.26 | 81.95±0.71 | 81.87±0.71 | 90.65±0.33 | 90.63±0.32 | 89.04±0.17 | 89.00±0.20 | 91.72±1.29 | 91.73±1.29 | 88.70±0.32 | 88.67±0.33 |
| FedAPT [AAAI 2024] | 91.36±0.11 | 91.33±0.15 | 85.25±0.87 | 85.70±0.28 | 90.40±0.73 | 90.37±0.76 | 90.68±0.48 | 90.63±0.51 | 94.95±0.30 | 94.93±0.32 | 90.53±0.37 | 90.98±0.46 |
| FOCoOP [ICML 2025] | 76.70±14.09 | 74.33±17.06 | 74.27±9.28 | 72.20±11.74 | 72.98±14.48 | 69.33±19.54 | 76.05±13.02 | 73.80±15.88 | 72.61±8.92 | 69.93±11.10 | 74.52±11.07 | 71.92±14.13 |
| FedMVP [ICCV 2025] | 89.87±3.10 | 89.87±3.09 | 82.10±0.83 | 81.83±0.87 | 89.74±4.84 | 89.73±4.88 | 90.80±2.19 | 90.80±2.17 | 91.55±5.01 | 91.53±5.06 | 88.81±3.10 | 88.75±3.09 |
| **FedMAP (Ours)** | **93.71±0.39** | **93.10±0.17** | **87.48±4.53** | **87.33±4.79** | **93.23±0.17** | **93.20±0.17** | **94.72±0.11** | **94.73±0.06** | **96.60±0.31** | **96.60±0.35** | **93.15±0.87** | **92.99±0.81** |

*Table 3.* Comparative results on the **Private** dataset. We report Accuracy/Macro-F1 (%) over three seeds for trainable methods; Zero-Shot CLIP is deterministic.

| Methods | RMD | | RMZ | | SFY | | Avg | |
|---|---|---|---|---|---|---|---|---|
| | Acc | F1 | Acc | F1 | Acc | F1 | Acc | F1 |
| Zero-Shot CLIP [ICML 2021] | 6.96 | 4.50 | 7.83 | 4.20 | 13.91 | 7.30 | 9.57 | 5.33 |
| PromptFL [TMC 2023] | 63.00±4.76 | 57.50±7.55 | 78.97±0.69 | 75.73±0.76 | 62.05±3.58 | 58.47±3.40 | 68.01±0.87 | 63.90±1.46 |
| PromptFL+Prox [PMLR 2020] | 63.00±4.76 | 57.50±7.55 | 78.97±0.69 | 75.73±0.76 | 62.05±3.58 | 58.47±3.40 | 68.01±0.87 | 63.90±1.46 |
| FedKgCoOP [CVPR 2023] | 58.85±2.94 | 53.03±4.43 | 71.05±1.71 | 66.50±1.39 | 58.21±1.85 | 54.17±2.20 | 62.70±1.41 | 57.90±1.55 |
| FedCLIP [ICLR 2023] | 54.33±1.06 | 49.03±2.40 | 71.37±5.60 | 66.80±5.91 | 59.03±5.33 | 56.60±6.08 | 61.58±3.99 | 57.48±4.78 |
| FedAPT [AAAI 2024] | 65.45±4.34 | 60.70±5.30 | 80.86±1.56 | 77.83±1.59 | 65.95±1.47 | 63.07±0.84 | 70.75±1.13 | 67.20±1.03 |
| FOCoOP [ICML 2025] | 35.90±12.82 | 27.43±14.73 | 37.22±11.13 | 31.33±9.83 | 31.85±7.68 | 25.93±8.86 | 34.99±10.52 | 28.23±11.08 |
| FedMVP [ICCV 2025] | 46.76±5.44 | 39.67±6.86 | 63.19±3.12 | 58.30±4.12 | 45.20±1.33 | 42.60±1.21 | 51.72±2.67 | 46.86±3.28 |
| **FedMAP (Ours)** | **87.67±2.96** | **85.70±3.81** | **94.55±0.00** | **92.80±0.35** | **86.57±1.02** | **85.63±1.33** | **89.60±1.33** | **88.04±1.83** |

### 3.5. Optimization Objective and Algorithm

The final objective function for each client $k$ integrates the manifold anchoring loss and the topological alignment regularization:

$$\mathcal{L}_{total} = \mathcal{L}_{CE}(\mathbf{v}, y; \mathbf{A}) + \lambda \cdot \mathcal{L}_{struct}, \qquad (9)$$

where $\lambda$ is a hyperparameter balancing the classification accuracy and geometric structural consistency. The training procedure of FedMAP is summarized in Algorithm 1.

## 4. Experiments

### 4.1. Experimental Setup

**Datasets and Protocols.** We evaluate on three medical federated benchmarks: **FedISIC** (dermoscopy) (Ogier du Terrail et al., 2022a), with 3 classes across 6 domains; **Fed-Camelyon17** (histopathology) (Bandi et al., 2018; Koh et al., 2021), with 2 classes across 5 hospitals; and a **Private Fetal Ultrasound** dataset with 13 classes across 3 hospitals. We adopt a **one-domain-one-client** protocol.

**Baselines.** We organize baselines into four groups to cover different adaptation regimes: (i) a no-update reference, Zero-shot CLIP (Radford et al., 2021); (ii) federated text-prompt tuning, PromptFL (Guo et al., 2023b) and PromptFL+Prox (Li et al., 2020b); (iii) lightweight CLIP/FPL adaptation under heterogeneity, FedCLIP (Lu et al., 2023), FedAPT (Su et al., 2024), and FOCoOP (Liao et al., 2025); and (iv) knowledge- or multimodal-enhanced prompt learning, FedKgCoOP (Yao et al., 2023) and FedMVP (Singha et al., 2025). All methods use the same frozen CLIP ViT-B/16 (Radford et al., 2021) backbone and follow official implementations.

**Implementation Details.** FedMAP inserts visual prompts into all 12 visual-transformer layers with prompt length 1; text-prompt baselines use context length 16. We train for 50 communication rounds with 1 local epoch per round, set $\lambda=10$, and report client-averaged Top-1 accuracy and Macro-F1. Unless otherwise noted, trainable methods are averaged over three seeds.

### 4.2. Main Results: Comparative Analysis

Table 1–Table 3 summarize results on dermoscopic, histopathological, and ultrasound medical benchmarks. FedMAP achieves the best Accuracy and Macro-F1.

*Table 4.* **Anchor construction ablation for MSA.** We compare alternative semantic anchor designs; **Random** and noisy variants test whether *meaningful visual semantics* drive the gain. **Avg** denotes the mean accuracy across datasets.

| Anchor Type | FedISIC | FedCamelyon17 | Private | Avg |
|---|---|---|---|---|
| Random | 57.05 | 90.83 | 40.13 | 62.67 |
| CLS-only | 70.04 | 93.08 | 89.25 | 84.12 |
| Template | 70.15 | 93.17 | 89.58 | 84.30 |
| Attribute-only | 71.05 | 93.43 | 90.08 | 84.85 |
| Single (one anchor) | 68.88 | 92.43 | 89.18 | 83.50 |
| **Gauss Noisy** | | | | |
| Low ($\sigma$=0.01) | 72.65 | 93.57 | 85.11 | 83.78 |
| Medium ($\sigma$=0.05) | 64.02 | 93.07 | 82.25 | 79.78 |
| High ($\sigma$=0.10) | 49.08 | 92.93 | 19.52 | 53.84 |
| **MSA (codebook ensemble)** | **73.23** | **93.64** | **90.36** | **85.74** |

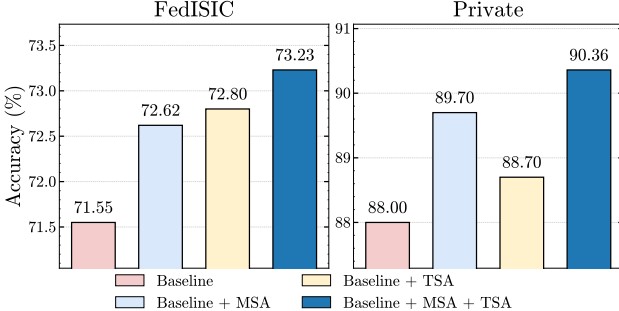

*Figure 5.* **Module ablation.** We evaluate **Baseline** (visual prompts only), **+MSA**, **+TSA**, and **+MSA+TSA**. Both modules are complementary, and their combination achieves the best performance across all benchmarks.

On **FedISIC**, FedMAP reaches **73.29±1.29%** accuracy and **59.47±1.98%** Macro-F1, improving over the strongest baselines by **+7.16%** accuracy and **+19.56%** Macro-F1. On **FedCamelyon17**, it obtains **93.15±0.87% / 92.99±0.81%**, surpassing FedAPT (Su et al., 2024) by **+2.62% / +2.01%**. On the **Private** dataset, FedMAP achieves **89.60±1.33% / 88.04±1.83%**, improving over FedAPT by **+18.85% / +20.84%**. These consistent gains support semantic anchoring and topology alignment for medical FPL.

The improvement is not driven only by average aggregation. Across the three benchmarks, FedMAP also shows **consistent domain-wise advantages**: it performs best on all hospitals of FedCamelyon17, all sites of the private ultrasound dataset, and remains the strongest or highly competitive across the FedISIC sites. This domain-wise behavior indicates that visual manifold anchoring reduces center-specific overfitting and improves cross-client robustness, rather than benefiting from a single favorable domain.

### 4.3. Ablation Studies

We conduct extensive ablation studies to examine FedMAP from multiple complementary angles, including module contributions, anchor design, structural alignment objectives, hyper-parameter sensitivity, prompt configuration, and communication efficiency.

*Table 5.* **TSA relation-distance ablation.** We compare alternative distance metrics for aligning the visual relation matrix $\mathbf{G}_{vis}$ with the text reference $\mathbf{G}_{text}$. Results are reported as average accuracy.

| Relation Distance | FedISIC | Private |
|---|---|---|
| KL divergence | 70.99 | 88.89 |
| 1−cosine similarity | 72.79 | 89.72 |
| Bures–Wasserstein | 72.17 | 89.43 |
| **MSE (Ours)** | **73.06** | **90.36** |

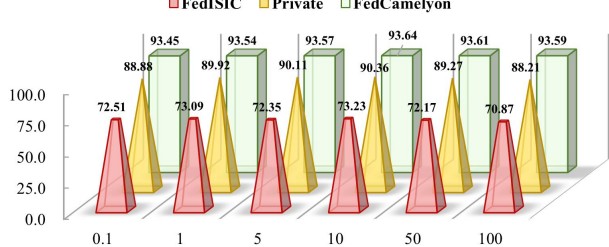

*Figure 6.* **Sensitivity to $\lambda$.** We sweep the structural loss weight across all three benchmarks. Performance is stable over a wide range and peaks around $\lambda$=10.

**Module ablation.** We evaluate *Baseline*, *+MSA*, *+TSA*, and *+MSA+TSA*. Figure 5 shows that both modules consistently improve over the baseline. On FedISIC, MSA and TSA improve accuracy from 71.55% to 72.62% and 72.80%, respectively; on the private dataset, they improve accuracy from 88.00% to 89.70% and 88.70%. Their combination further reaches 73.23% and 90.36%, showing that semantic anchoring and topology alignment address complementary failure modes rather than duplicating the same regularization effect.

**Anchor construction.** We compare six anchor designs: class-name, template, attribute-only, noisy, single-anchor, and full codebook. Table 4 shows a clear semantic-quality trend: random anchors underperform all meaningful textual anchors, while the full codebook achieves the best average accuracy (85.74%). Adding Gaussian noise progressively degrades performance, with $\sigma$=0.10 causing a severe drop. These results show that FedMAP benefits from clean and diverse visual semantics rather than arbitrary classifier directions, and that ensembling complementary attributes reduces sensitivity to individual LLM prompt wording.

**Topology alignment.** Replacing MSE with KL, cosine-style matching, or Bures–Wasserstein distance lowers accuracy on both evaluated datasets (Table 5). MSE reaches 73.06% on FedISIC and 90.36% on the private dataset, outperforming the closest alternatives. This suggests that directly matching dense class-relation matrices provides a stable structural constraint. The advantage of MSE is expected: $\mathbf{G}_{text}$ encodes pairwise similarity scores rather than calibrated probability distributions, making regression-style matching more natural than distributional alternatives such

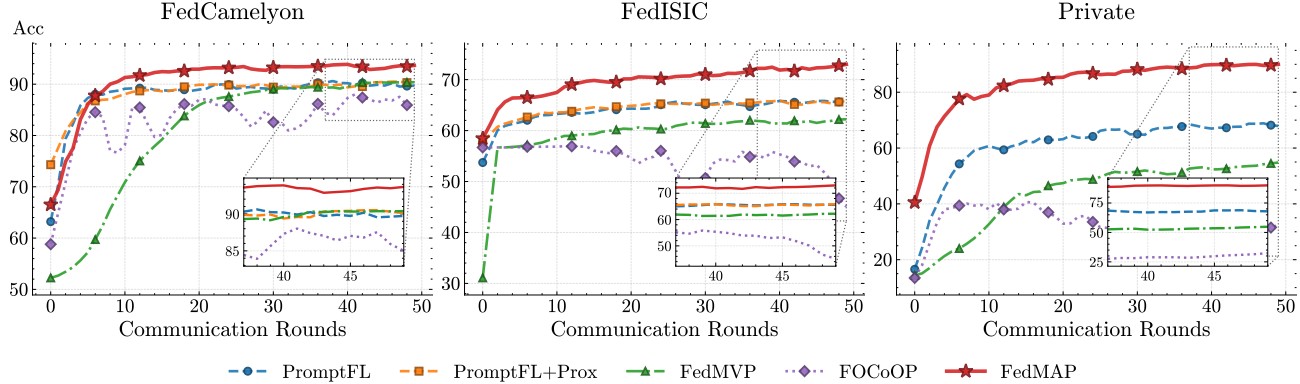

*Figure 7.* **Convergence curves.** Test accuracy across communication rounds. FedMAP converges faster and reaches a higher final accuracy than PromptFL, PromptFL+Prox, FedMVP, and FOCoOP on all three benchmarks.

*Table 6.* **Parameter efficiency and communication cost.** FedMAP uses only **9.2K** trainable parameters and **0.07 MB** per round, making it the lightest among the shown visual/multimodal adaptation methods. Communication is measured in FP16 MB.

| Method | Params (M) | Round (MB) | Total (MB) |
|---|---|---|---|
| FedAPT [AAAI 2024] | 0.0113 | 0.23 | 11.53 |
| FOCoOP [ICML 2025] | 0.1475 | 0.56 | 28.13 |
| FedMVP [ICCV 2025] | 9.8496 | 37.57 | 1878.66 |
| **FedMAP (ours)** | **0.0092** | **0.07** | **3.34** |

as KL divergence, especially under non-IID federated optimization where clients exhibit varying score scales.

**Hyper-parameter sensitivity.** FedMAP is stable across $\lambda$ and EMA momentum $\mu$ sweeps (Figures 6 and 8). A very small $\lambda$ underuses the topology constraint, while an overly large value can over-constrain the local visual prototypes; performance peaks around $\lambda=10$ across the three benchmarks. The EMA study shows the same pattern: a high momentum gives the most reliable prototype bank by filtering minibatch noise while preserving cross-round memory. We therefore use $\lambda=10$ and $\mu=0.9$ by default.

**Prompt hyper-parameters.** Depth is more influential than length once sufficient prompt capacity is provided (Figure 9). Increasing visual prompt depth lets FedMAP correct feature geometry across more transformer layers, which is important for reversing medical manifold collapse. By comparison, increasing prompt length mainly adds local capacity and quickly saturates. This supports the default design of using deep but lightweight visual prompts.

**Efficiency and scalability.** Table 6 shows that FedMAP uses only **9.2K** trainable parameters and **0.07 MB** per client per round, yielding the lowest communication cost among the methods reported in the table. In particular, FedMAP is far lighter than FedMVP while obtaining stronger main results, which indicates that the improvement is not obtained by expanding the trainable model size. Split-client scalability in Appendix Tables 9 to 11 further shows that the method remains practical when each original domain is divided into

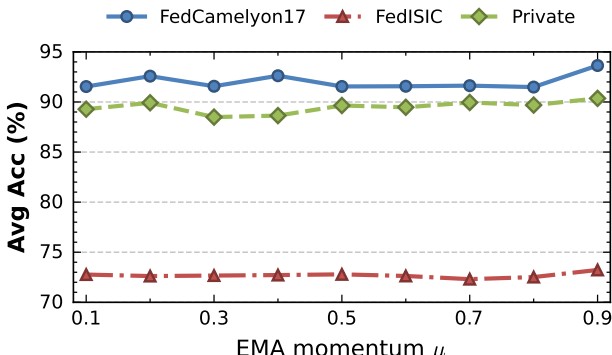

*Figure 8.* **EMA momentum sensitivity.** We sweep the prototype EMA momentum $\mu$. Higher values ($\mu \geq 0.9$) yield more stable prototype estimates by filtering minibatch noise.

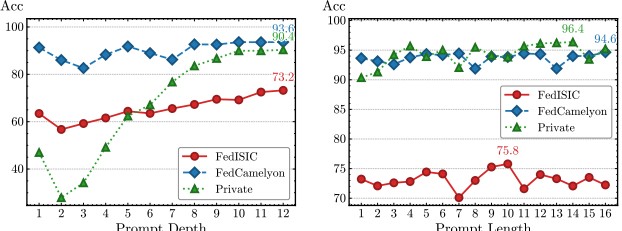

*Figure 9.* **Prompt hyper-parameters.** We study the effect of visual prompt **depth** (number of prompted layers) and **length** (number of tokens per layer). Depth is more influential than length for reversing medical manifold collapse.

more clients.

**Convergence.** FedMAP converges faster and reaches higher final accuracy than PromptFL, PromptFL+Prox, Fed-MVP, and FOCoOP (Figure 7). The advantage appears throughout training rather than only at the final round: FedMAP rises quickly in early communication rounds and maintains a higher plateau on all three datasets. This behavior is consistent with the role of the fixed semantic anchors and the EMA prototype bank, which provide a stable optimization reference under client heterogeneity. In contrast, methods relying solely on text-side tuning (PromptFL, PromptFL+Prox) or lacking an explicit structural reference (FOCoOP, FedMVP) exhibit slower convergence and greater

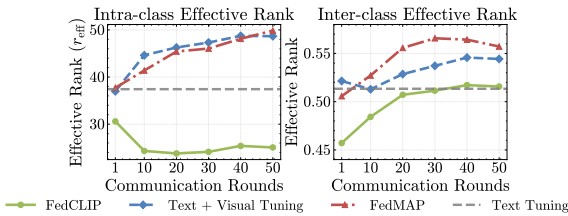
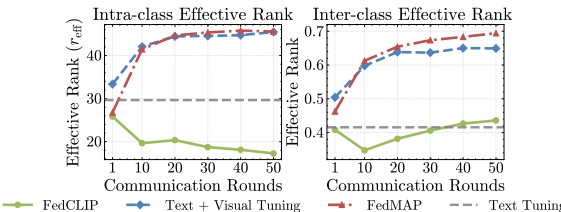

*(a)* FedISIC          *(b)* Private

*Figure 10.* **Effective-rank evolution.** Intra-class effective rank across communication rounds on FedISIC and Private. FedMAP consistently expands the feature subspace, while baselines remain stagnant or degrade.

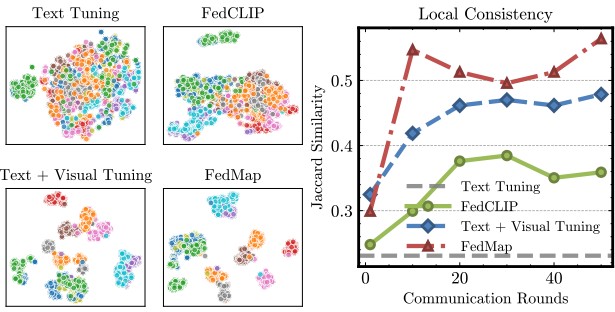

*Figure 11.* **Private dataset: t-SNE** visualization (left) and **Jaccard Similarity** (right) comparing embedding geometry across methods. FedMAP yields higher local topology stability.

variance across rounds, particularly on FedISIC where non-IID severity is highest.

### 4.4. In-depth Analysis

**Effective-rank expansion.** To quantify how well FedMAP reverses manifold collapse, we track the **intra-class effective rank** across communication rounds. Given a feature matrix $V \in \mathbb{R}^{N \times D}$, let $\Sigma(V)$ be its centered covariance matrix with eigenvalues $\lambda_1 \geq \cdots \geq \lambda_R$, where $R = \min\{N - 1, D\}$. Defining $p_i = \lambda_i / \sum_{j=1}^{R} \lambda_j$, the effective rank is the exponential of the spectral entropy:

$$r_{eff}(V) = \exp\Big(-\sum_{i=1}^{R} p_i \ln p_i\Big). \quad (10)$$

Intuitively, $r_{eff}$ measures the intrinsic dimensionality of the feature distribution—a lower value indicates a more collapsed manifold. For class $c$ with feature matrix $V_c \in \mathbb{R}^{N_c \times D}$, we compute $r_{eff}(V_c)$ per class and report the macro average:

$$r_{\text{intra}} = \frac{1}{C} \sum_{c=1}^{C} r_{eff}(V_c). \quad (11)$$

Similarly, stacking class-mean features as $\bar{V} = [\bar{\mathbf{v}}_1; \ldots; \bar{\mathbf{v}}_C] \in \mathbb{R}^{C \times D}$, we define the **inter-class effective rank** as $r_{\text{inter}} = r_{eff}(\bar{V})$ and use $r_{\text{inter}}/(C - 1)$ for cross-dataset comparisons, complementing the analysis in Figure 2. As shown in Figure 10, FedMAP consistently expands the intra-class effective rank on FedISIC and Private throughout training, while baseline methods remain stagnant

or degrade. This confirms that visual manifold anchoring progressively recovers discriminative degrees of freedom that are suppressed in the original frozen features.

**t-SNE visualization.** We compare the learned embedding geometry of different methods on the Private dataset at the final communication round. As shown in Figure 11 (left), FedMAP produces more compact and better-separated t-SNE clusters than Text Tuning (**Static**), FedCLIP, and Text+Visual Tuning.

To quantify local topology stability, we measure **Jaccard Similarity** between consecutive communication rounds. For each sample $x$, let $\mathcal{N}_2^{(t)}(x)$ be its 2-nearest-neighbor set at round $t$. The neighborhood overlap between rounds $t - 1$ and $t$ is:

$$\mathcal{J}(x) = \frac{|\mathcal{N}_2^{(t)}(x) \cap \mathcal{N}_2^{(t-1)}(x)|}{|\mathcal{N}_2^{(t)}(x) \cup \mathcal{N}_2^{(t-1)}(x)|}. \quad (12)$$

Higher Jaccard values indicate more stable local structure under federated aggregation. As shown in Figure 11 (right), FedMAP achieves the highest and most consistently increasing Jaccard Similarity across rounds, substantially above all baselines. This indicates that MSA and TSA jointly stabilize the neighborhood topology throughout training, suggesting that client updates are better aligned to a shared semantic geometry and thus yield stronger generalization.

## 5. Conclusion

In this paper, we revisit federated prompt learning from a geometric perspective and identify two key failure modes in medical imaging: *intra-client manifold collapse* and *inter-client topological misalignment*. To address them, we propose **FedMAP**, which shifts FPL beyond textual tuning via **visual manifold anchoring** with lightweight visual prompts and LLM-derived semantic regularization. Experiments on FedISIC, FedCamelyon17, and a private ultrasound benchmark demonstrate consistent gains in performance and convergence stability under heterogeneous clients. Future work includes improving the robustness of anchor construction, extending the framework to broader modalities and supervision regimes, and exploring iterative codebook refinement with complementary clinical modalities when available.

## Impact Statement

This work aims to make federated adaptation of medical foundation models more practical, privacy-preserving, and interpretable. By identifying intra-client manifold collapse and inter-client topological misalignment as key failure modes, FedMAP offers the community a geometric lens for diagnosing federated prompt learning beyond average accuracy. Its lightweight visual manifold anchoring may support future multi-center applications in dermatology, pathology, ultrasound, and other imaging workflows where raw data cannot be shared and communication budgets are limited. At the same time, real-world use must account for dataset shift, subgroup bias, and possible errors in automatically generated semantic attributes. FedMAP keeps patient data local and uses the LLM-derived codebook only as a semantic reference, but deployment still requires subgroup evaluation, cross-site auditing, clinical expert review when applicable, and continuous monitoring; it should not be used as a standalone clinical decision system without proper validation and oversight.

## Acknowledgements

This work was supported in part by the National Key Research and Development Program of China (2023YFC2705701), the National Natural Science Foundation of China (62225113), the Innovative Research Group Project of Hubei Province (2024AFA017), the New Cornerstone Science Foundation through the XPLORER PRIZE, the National Key Research and Development Program of China (2026YFE0202100), the National Natural Science Foundation of China under Grant (62361166629), the Major Project of Science and Technology Innovation of Hubei Province (2024BCA003), the Key Research and Development Program of Wuhan (2025061202030423), and the National Natural Science Foundation of China (62506269, 623B2080).

This work was also supported by WHU-Kingsoft Joint Lab and the NTU AI-for-X Postdoctoral Fellowship. The numerical calculations in this paper were supported by the supercomputing system at the Supercomputing Center of Wuhan University.

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

# A. Method Details

## A.1. Notation Table

To facilitate understanding of our proposed FedMAP, we provide a summary of key notations used throughout this paper in Table 7.

*Table 7.* **Notation** table used in FedMAP.

| Symbol | Description | Symbol | Description |
|---|---|---|---|
| $K$ | Number of clients | $C$ | Number of classes |
| $k$ | Client index | $\mathcal{D}_k$ | Local dataset at client $k$ |
| $N_k$ | Samples at client $k$ | $N$ | Total samples ($N = \sum_k N_k$) |
| $\mathcal{X}$ | Input image space | $\mathcal{Y}$ | Label set ($\{1, \ldots, C\}$) |
| $\Theta_{frozen}$ | Frozen CLIP backbone parameters | $\mathcal{E}_I$ | Frozen image encoder |
| $\mathcal{E}_T$ | Frozen text encoder | $\mathbf{P}$ | Visual prompts (all prompted layers) |
| $\mathbf{P}^{(\ell)}$ | Prompt tokens at layer $\ell$ | $\mathcal{L}_p$ | Set of prompted layers |
| $\mathbf{P}_g$ | Server/global prompt in a round | $\mathbf{P}_k$ | Client-$k$ local prompt |
| $\mathbf{v}$ | Visual feature embedding | $\tau$ | Temperature in softmax |
| $\mathcal{S}_c$ | Attribute set for class $c$ | $M$ | Number of attributes per class |
| $\phi(c, s)$ | Class-attribute prompt template | $\mathbf{a}_c$ | Semantic anchor for class $c$ |
| $\mathbf{A}$ | Semantic anchor matrix | $\mathbf{G}_{text}$ | Text relation matrix |
| $\mathbf{Q}$ | Prototype bank ($\{\mathbf{q}_c\}_{c=1}^C$) | $\mathbf{q}_c$ | Class prototype of class $c$ |
| $\mathbf{G}_{vis}$ | Visual relation matrix (prototype Gram) | $\mathcal{L}_{CE}$ | Classification loss (MSA) |
| $\mathcal{L}_{struct}$ | Structural loss (TSA) | $\lambda$ | Weight of $\mathcal{L}_{struct}$ |
| $\mu$ | EMA momentum | $\eta$ | Learning rate |
| $t$ | Communication round index | $E$ | Local epochs per round |
| $T$ | Total communication rounds | $m_p$ | Visual prompt length |

## A.2. Algorithm of FedMAP

The complete training procedure of FedMAP is summarized in Algorithm 1.

## A.3. LLM-Based Medical Codebook Construction

To improve reproducibility, we detail the multi-turn prompting strategy used to construct the Semantic Codebook via **GPT-4o** (Hurst et al., 2024). We adopt a "describe-then-abstract" prompting strategy to distill high-level semantic dimensions from specific observational cues.

**System Instruction.** The LLM is initialized with the following system role:

> *"You are a medical domain expert and prompt engineer. Follow the format strictly. No extra commentary."*

**Round 1: Observational Description.** In the first turn, we guide the LLM to generate concise visual observations for each class, strictly limiting the output to observable cues to avoid hallucinations or non-visual medical knowledge.

**User Prompt:**
"I have the following medical classes:
[MEDICAL_CLASS_LIST]
For each class, write one short observational sentence
($\leq 15$ words).
Rules:
1. Mention only observable cues (anatomy/site, modality if relevant, pattern, color/signal, border, surface, vessels).
2. No explanations, no "coverage/observability", no

---

**Algorithm 1** FedMAP: Federated Manifold-Anchored Prompting

**Input:** Frozen CLIP backbone $\Theta_{frozen}$ (incl. $\mathcal{E}_I, \mathcal{E}_T$), Client datasets $\{\mathcal{D}_k\}_{k=1}^K$,
Comm. rounds $T$, Local epochs $E$, Attributes per class $M$, Hyperparams $\eta, \tau, \mu, \lambda$
**Output:** The global visual prompt $\mathbf{P}_g^T$, Semantic Anchors $\mathbf{A}$

/* Server-side: Semantic Codebook Construction (Round 0) */
Generate attribute sets $\{\mathcal{S}_c\}_{c=1}^C$ via LLM Section 3.3
Construct Anchor Matrix $\mathbf{A} = [\mathbf{a}_1, \ldots, \mathbf{a}_C]$ via Equation (3)
Compute Text Relation Matrix $\mathbf{G}_{text} = \mathbf{A}^\top \mathbf{A}$ via Equation (5)
Initialize global visual prompt $\mathbf{P}_g^0$

/* Federated Visual Manifold Anchoring */
**for** $t = 0, 1, \ldots, T - 1$ **do**
  Server broadcasts global prompt $\mathbf{P}_g^t$, Anchors $\mathbf{A}$, and Relation $\mathbf{G}_{text}$
  **for** client $k \in \{1, \ldots, K\}$ **in parallel do**
    $\mathbf{P}_k^t \leftarrow \mathbf{P}_g^t$ ;    // Init local prompt
    Initialize/Load local prototype bank $\mathbf{Q}$
    **for** $e = 1, \ldots, E$ **do**
      **for** $(x, y) \in \mathcal{D}_k$ **do**
        /* Manifold Semantic Anchoring */
        $\mathbf{v} = \mathcal{E}_I([x, \mathbf{P}_k^t]; \Theta_{frozen})$
        $\mathcal{L}_{CE} \leftarrow -\log p(y \mid \mathbf{v}; \mathbf{A})$ via Equation (4)
        /* Topology Structural Alignment */
        Update observed class prototypes in $\mathbf{Q}$ via Equation (6)
        $\mathbf{G}_{vis} = \mathbf{Q}^\top \mathbf{Q}$ via Equation (7)
        $\mathcal{L}_{struct} = \|\mathbf{G}_{vis} - \mathbf{G}_{text}\|_F^2$ via Equation (8)
        /* Optimization */
        $\mathcal{L}_{total} = \mathcal{L}_{CE} + \lambda \mathcal{L}_{struct}$ via Equation (9)
        $\mathbf{P}_k^t \leftarrow \mathbf{P}_k^t - \eta \nabla_{\mathbf{P}_k^t} \mathcal{L}_{total}$
      **end**
    **end**
    Client uploads $\mathbf{P}_k^t$ to server
  **end**
  /* Server Aggregation */
  $\mathbf{P}_g^{t+1} \leftarrow \sum_{k=1}^K \frac{|\mathcal{D}_k|}{|\mathcal{D}|} \mathbf{P}_k^t$
**end**
**return** $\mathbf{P}_g^T, \mathbf{A}$

---

advice.
Output format exactly:
<class>: <one short sentence>"

**Round 2: Semantic Dimension Abstraction.** In the second turn, we feed the observational sentences back to the model to extract abstract semantic dimensions. This ensures the anchors are generalizable rather than overly specific sentences.

**User Prompt:**
"From the sentences above, output exactly 10 semantic anchors for prompt learning.
Hard constraints (must be satisfied):

1. Output ONLY 10 lines.
2. Each line is 1–2 words (English), Title Case.
3. No punctuation, no numbering, no bullets, no explanations.
4. Each anchor must represent one attribute dimension (not a full concept description).
5. Use clinical-appropriate general dimensions, not dataset-specific jargon.
If you output anything else, retry silently and comply."

**Anchor Template Construction.**  Finally, as described in Section 3.3, these extracted semantic dimensions $s_{c,k}$ are combined with the class name $c$ using the template "`{cls}:{attribute}`" before being encoded by the frozen text encoder. This two-step process ensures that the anchors $\mathbf{A}$ capture the distinct geometric directions relevant to the medical task.

**Example Codebook Snippet.**  Example generated visual attributes:

- "homogeneous hemorrhagic blotches"
- "clustered round or oval globules"
- "well circumscribed red to purple areas"
- "dark surface crusting"
- "round to oval lacunae"
- "dotted pinpoint vessels"
- "pale whitish septa"
- "variable vessel density"
- "dusky blue shadowing"
- "linear irregular serpentine vessels"

## A.4. Theoretical Analysis.

### A.4.1. THEOREMS

For compact notation in this theoretical analysis, $P$ denotes the vectorized visual prompt parameters $\mathbf{P}$ used in the main text.

**Theorem A.1** (Why Textual Tuning Breaks, and Why MSA/TSA Fix It). *Fix a client $k$ with local distribution $\mathcal{D}_k$ over $(x, y) \in \mathcal{X} \times \mathcal{Y}$, $\mathcal{Y} = \{1, \ldots, C\}$. Let the frozen image encoder be $E_I(\cdot; \Theta^{\text{frozen}})$ and the learnable visual prompt be $P$, producing*

$$v = E_I([x, P]; \Theta^{\text{frozen}}) \in \mathbb{R}^d. \quad (13)$$

*Let semantic anchors be $\{a_c\}_{c=1}^C$ constructed by Equation (3), and stack them as $A = [a_1, \ldots, a_C] \in \mathbb{R}^{d \times C}$.*

*The MSA classifier uses the anchor-based distribution Equation (4):*

$$p(y = c \mid x) = \frac{\exp(\text{sim}(v, a_c)/\tau)}{\sum_{j=1}^C \exp(\text{sim}(v, a_j)/\tau)}. \quad (14)$$

$$\mathcal{L}_{\text{CE}}(v, y; A) = -\log p(y \mid x). \quad (15)$$

*For TSA, each client maintains prototypes $\{q_c\}_{c=1}^C$ (Equation (6)) stacked as $Q = [q_1, \ldots, q_C] \in \mathbb{R}^{d \times C}$, and defines*

$$G^{\text{text}} = A^\top A. \quad (16)$$

$$G^{\text{vis}} = Q^\top Q. \quad (17)$$

$$\mathcal{L}_{\text{struct}} = \|G^{\text{vis}} - G^{\text{text}}\|_F^2. \quad (18)$$

**Textual tuning limitation under manifold collapse.**  *Assume class-wise manifold collapse on client $k$: for each class $c$,*

$$v \mid (y = c) = \mu_{k,c} + U_{k,c}z,$$
$$z \text{ is zero-mean}, \quad (19)$$
$$r_{k,c} \ll d.$$

*with $U_{k,c} \in \mathbb{R}^{d \times r_{k,c}}$ and define the effective subspace*

$$S_k = \text{span}\Big(\{\mu_{k,c}\}_{c=1}^C \cup \bigcup_{c=1}^C \text{col}(U_{k,c})\Big), \quad (20)$$

*$P_{S_k}$ denotes the orthogonal projector onto $S_k$.*

*Consider any textual tuning rule that adapts text-side classifier weights $\{t_c\}_{c=1}^C \subset \mathbb{R}^d$ while keeping $v$ fixed, and predicts by logits $\ell_c(v) = \text{sim}(v, t_c)/\tau$. Then for any pair $c \neq c'$, the class separation is fully determined by the projected discrepancy in $S_k$:*

$$\ell_c(v) - \ell_{c'}(v) = \frac{1}{\tau}\Big\langle P_{S_k}v, \; P_{S_k}(t_c - t_{c'})\Big\rangle,$$

$$\Rightarrow \sup_{\{t_c\}: \; \|P_{S_k}(t_c - t_{c'})\|_2 \leq 1} \Big|\ell_c(v) - \ell_{c'}(v)\Big| \leq \frac{1}{\tau}\|P_{S_k}v\|_2. \quad (21)$$

*In particular, no textual tuning can create discriminative directions in $S_k^\perp$ because $v$ has zero variation there under Equation (19)–Equation (20).*

**(ii) MSA provides anchor-span descent directions.**  *Let $\pi(v) \in \mathbb{R}^C$ be the softmax probabilities in Equation (14), and let $e_y \in \mathbb{R}^C$ be the one-hot label vector. Assuming $\text{sim}(v, a) = v^\top a$, the gradient of $\mathcal{L}_{\text{CE}}$ w.r.t. $v$ satisfies*

$$\nabla_v \mathcal{L}_{\text{CE}}(v, y; A) = \frac{1}{\tau}A\big(\pi(v) - e_y\big) \in \text{span}(A). \quad (22)$$

*Consequently, the prompt update $\nabla_P \mathcal{L}_{\text{CE}} = J_P(v)^\top \nabla_v \mathcal{L}_{\text{CE}}$ is always driven by semantic anchor directions, i.e., MSA injects a client-invariant geometric "blueprint" through $\text{span}(A)$.*

**(iii) TSA enforces inter-class topology consistency.** *Let* $\Delta \triangleq G^{vis} - G^{text}$. *Then entrywise similarity mismatch is controlled by the structural loss:*

$$
\begin{aligned}
\max_{i,j \in \{1,\dots,C\}} \left| q_i^\top q_j - a_i^\top a_j \right| &= \max_{i,j} |\Delta_{ij}| \\
&\leq \|\Delta\|_F = \sqrt{\mathcal{L}_{struct}}.
\end{aligned} \tag{23}
$$

*Moreover, define an anchor-side neighborhood margin (for a fixed neighborhood size $m$)*

$$
\gamma_m \triangleq \min_i \left( (a_i^\top a_{(m)}^{(i)}) - (a_i^\top a_{(m+1)}^{(i)}) \right), \tag{24}
$$

*where $a_{(m)}^{(i)}$ denotes the $m$-th nearest anchor to $a_i$ under dot-product similarity. If $\sqrt{\mathcal{L}_{struct}} < \gamma_m/2$, then the top-$m$ neighbor set of each class prototype $\{q_c\}$ matches that of anchors $\{a_c\}$, hence TSA yields neighborhood/topology alignment across clients.*

**Explanation.** Theorem A.1 formalizes three phenomena observed in medical FPL: (i) when the frozen visual features collapse into a low-dimensional client-specific subspace $S_k$, text-side boundary shifting is intrinsically ill-conditioned because it cannot invent discriminative directions outside $S_k$; (ii) MSA anchors the optimization to a fixed semantic span $\mathrm{span}(A)$ by Equation (22), which provides stable geometric directions to counteract rank deficiency; (iii) TSA aligns the *pairwise* class similarity structure by matching Gram matrices, and Equation (23)–Equation (24) shows that small $\mathcal{L}_{struct}$ implies neighbor-level topology consistency.

**Theorem A.2** (FedMAP Convergence). *Define the client objective as*

$$
F_k(P) = \mathbb{E}_{(x,y)\sim\mathcal{D}_k}\left[\mathcal{L}_{CE}(v,y;A)\right] + \lambda \mathcal{L}_{struct,k}, \tag{25}
$$

$$
F(P) = \sum_{k=1}^{K} \frac{N_k}{N} F_k(P), \tag{26}
$$

*where $v = E_I([x,P];\Theta^{frozen})$ and $\mathcal{L}_{struct,k}$ is the TSA loss computed on client $k$, possibly on observed class subsets. Assume:*

- *(Smoothness) $F_k$ is $L$-smooth for all $k$.*

- *(Stochastic gradients) Each client uses unbiased stochastic gradients with variance at most $\sigma^2$.*

- *(Client drift) Define heterogeneity $\mathcal{H}(P) \triangleq \frac{1}{K}\sum_{k=1}^K \|\nabla F_k(P) - \nabla F(P)\|_2^2$.*

*Run FedAvg on prompt parameters with step size $\eta$ and $E$ local steps per round, producing global prompts $\{P_g^t\}_{t=0}^{T-1}$.*

*Then for $\eta$ sufficiently small (standard FedAvg condition), the average stationarity satisfies*

$$
\begin{aligned}
\frac{1}{T}\sum_{t=0}^{T-1} \mathbb{E}\left[\|\nabla F(P_g^t)\|_2^2\right] &\leq \frac{2\big(F(P_g^0) - F^\star\big)}{\eta E T} \\
&+ \eta L\sigma^2 + \eta^2 L^2 E^2 \cdot \frac{1}{T}\sum_{t=0}^{T-1}\mathbb{E}\big[\mathcal{H}(P_g^t)\big].
\end{aligned} \tag{27}
$$

*Furthermore, the drift decomposes as*

$$
\mathcal{H}(P) \leq 2\mathcal{H}_{CE}(P) + 2\lambda^2 \mathcal{H}_{struct}(P), \tag{28}
$$

*and whenever TSA drives the structural discrepancy $\Delta_k \triangleq G_k^{vis} - G^{text}$ small on each client, the structural drift term is controlled:*

$$
\mathcal{H}_{struct}(P) \lesssim \frac{1}{K}\sum_{k=1}^{K}\|\Delta_k\|_F^2 = \frac{1}{K}\sum_{k=1}^{K}\mathcal{L}_{struct,k}. \tag{29}
$$

*Hence, by reducing $\sum_k \mathcal{L}_{struct,k}$, TSA reduces client-to-client geometric drift, leading to more stable aggregation and faster convergence in Equation (27).*

**Explanation.** Theorem A.2 uses a standard nonconvex FedAvg stationarity bound: the last term is the dominant "instability" source under non-IID data. Equation (29) states that TSA shrinks a part of this instability by enforcing a shared, client-invariant relational reference $G^{text}$.

A.4.2. PROOF

**Proof of Theorem A.1.**

**(i) Textual tuning limitation.** Under Equation (19)–Equation (20), we have $v \in S_k$ almost surely given any class label, hence $P_{S_k^\perp}v = 0$ and $v = P_{S_k}v$. For any text weights $\{t_c\}$, decompose $t_c = P_{S_k}t_c + P_{S_k^\perp}t_c$. Using $\mathrm{sim}(v,t) = v^\top t$ and $P_{S_k^\perp}v = 0$,

$$
\ell_c(v) = \frac{1}{\tau}v^\top t_c = \frac{1}{\tau}v^\top P_{S_k}t_c. \tag{30}
$$

Therefore, for $c \neq c'$,

$$
\ell_c(v)-\ell_{c'}(v) = \frac{1}{\tau}v^\top P_{S_k}(t_c-t_{c'}) = \frac{1}{\tau}\langle P_{S_k}v, \ P_{S_k}(t_c-t_{c'})\rangle, \tag{31}
$$

which proves Equation (21). Since $v$ has no variation in $S_k^\perp$, any discriminative direction must lie in $S_k$; thus the achievable separation is governed by the projected class discrepancy in $S_k$, i.e., by $\|P_{S_k}(\mu_{k,c} - \mu_{k,c'})\|$.

**(ii) MSA anchor-span gradients.** Let $s_c(v) \triangleq v^\top a_c/\tau$ and $\pi_c(v) = \exp(s_c(v))/\sum_j \exp(s_j(v))$. Then

$\mathcal{L}_{\text{CE}}(v, y; A) = -\log \pi_y(v)$, and by standard softmax calculus,

$$
\begin{aligned}
\nabla_v \mathcal{L}_{\text{CE}}(v, y; A) &= \sum_{c=1}^{C} \big( \pi_c(v) - \mathbf{1}[c = y] \big) \nabla_v s_c(v) \\
&= \frac{1}{\tau} \sum_{c=1}^{C} \big( \pi_c(v) - \mathbf{1}[c = y] \big) a_c \\
&= \frac{1}{\tau} A \big( \pi(v) - e_y \big).
\end{aligned}
\tag{32}
$$

which yields Equation (22) and $\nabla_v \mathcal{L}_{\text{CE}} \in \text{span}(A)$.

**(iii) TSA entrywise and neighborhood consistency.** By definition $\mathcal{L}_{\text{struct}} = \|\Delta\|_F^2$ with $\Delta_{ij} = q_i^\top q_j - a_i^\top a_j$. Since $\max_{i,j} |\Delta_{ij}| \le \|\Delta\|_F$, we get Equation (23). For the neighborhood claim, the assumption $\sqrt{\mathcal{L}_{\text{struct}}} < \gamma_m/2$ implies

$$
\big| q_i^\top q_j - a_i^\top a_j \big| < \gamma_m/2, \ \forall i, j,
\tag{33}
$$

so the similarity ordering around each $i$ cannot swap between the $m$-th and $(m+1)$-th neighbors (the anchor-side gap is $\gamma_m$), hence the top-$m$ neighbor set is preserved.

**Proof of Theorem A.2.**

**(A) Standard FedAvg stationarity bound.** Let the FedAvg round update be written as

$$
\begin{aligned}
P_g^{t+1} &= P_g^t - \eta E \, \bar{g}_t, \\
\bar{g}_t &\triangleq \frac{1}{E} \sum_{k=1}^{K} \alpha_k \sum_{e=0}^{E-1} \widehat{\nabla} F_k \big( P_k^{t,e} \big).
\end{aligned}
\tag{34}
$$

where $P_k^{t,0} = P_g^t$ and $P_k^{t,e+1} = P_k^{t,e} - \eta \widehat{\nabla} F_k(P_k^{t,e})$.

By $L$-smoothness of $F$, we have the one-step descent inequality

$$
F(P_g^{t+1}) \le F(P_g^t) - \eta E \big\langle \nabla F(P_g^t), \bar{g}_t \big\rangle + \frac{L \eta^2 E^2}{2} \|\bar{g}_t\|_2^2.
\tag{35}
$$

Decompose $\bar{g}_t$ as

$$
\begin{aligned}
\bar{g}_t &= \nabla F(P_g^t) + \delta_t + \varepsilon_t, \\
\mathbb{E}[\varepsilon_t \mid P_g^t] &= 0, \\
\mathbb{E}[\|\varepsilon_t\|_2^2] &\le \sigma^2.
\end{aligned}
\tag{36}
$$

where $\delta_t$ denotes the local-update drift. Using $\langle a, a+b \rangle \ge \frac{1}{2}\|a\|_2^2 - \frac{1}{2}\|b\|_2^2$ and $\|a+b+c\|_2^2 \le 3(\|a\|_2^2 + \|b\|_2^2 + \|c\|_2^2)$, (35) implies

$$
\begin{aligned}
\mathbb{E}[F(P_g^{t+1})] &\le \mathbb{E}[F(P_g^t)] - \frac{\eta E}{4} \mathbb{E}[\|\nabla F(P_g^t)\|_2^2] \\
&\quad + c_1 \, \eta E \, \mathbb{E}[\|\delta_t\|_2^2] + c_2 \, \eta^2 E^2 \, \sigma^2.
\end{aligned}
\tag{37}
$$

for universal constants $c_1, c_2 > 0$ (standard).

Under the standard FedAvg drift control (from $L$-smoothness and $E$ local steps),

$$
\begin{aligned}
\mathbb{E}[\|\delta_t\|_2^2] &\le L^2 E^2 \, \mathbb{E}[\mathcal{H}(P_g^t)], \\
\mathcal{H}(P_g^t) &\triangleq \frac{1}{K} \sum_{k=1}^{K} \big\| \nabla F_k(P_g^t) - \nabla F(P_g^t) \big\|_2^2.
\end{aligned}
\tag{38}
$$

which together with (37) yields (after telescoping over $t = 0, \ldots, T-1$ and using $F(P_g^T) \ge F^\star$)

$$
\begin{aligned}
\frac{1}{T} \sum_{t=0}^{T-1} \mathbb{E}[\|\nabla F(P_g^t)\|_2^2] &\le \frac{2 \big( F(P_g^0) - F^\star \big)}{\eta E T} \\
&\quad + \eta L \sigma^2 + \eta^2 L^2 E^2 \cdot \frac{1}{T} \sum_{t=0}^{T-1} \mathbb{E}[\mathcal{H}(P_g^t)].
\end{aligned}
\tag{39}
$$

which is Equation (27).

**(B) Drift decomposition for the composite objective.** Write $F_k(P) = F_{k,\text{CE}}(P) + \lambda F_{k,\text{struct}}(P)$ and $F(P) = \sum_k \alpha_k F_k(P)$. Then

$$
\begin{aligned}
\nabla F_k(P) - \nabla F(P) &= \big( \nabla F_{k,\text{CE}}(P) - \nabla F_{\text{CE}}(P) \big) \\
&\quad + \lambda \big( \nabla F_{k,\text{struct}}(P) - \nabla F_{\text{struct}}(P) \big).
\end{aligned}
\tag{40}
$$

and by $\|a + b\|_2^2 \le 2\|a\|_2^2 + 2\|b\|_2^2$,

$$
\mathcal{H}(P) \le 2 \mathcal{H}_{\text{CE}}(P) + 2\lambda^2 \mathcal{H}_{\text{struct}}(P),
\tag{41}
$$

which is Equation (28).

**(C) TSA controls the structural drift term.** By definition,

$$
\mathcal{L}_{\text{struct},k} = \|G_k^{\text{vis}} - G^{\text{text}}\|_F^2 = \|\Delta_k\|_F^2.
\tag{42}
$$

Assuming the structural gradient is Lipschitz in $\Delta_k$ (a standard regularity condition),

$$
\big\| \nabla F_{k,\text{struct}}(P) - \nabla F_{\text{struct}}(P) \big\|_2 \le c \, \|\Delta_k\|_F,
\tag{43}
$$

we obtain

$$
\begin{aligned}
\mathcal{H}_{\text{struct}}(P) &= \frac{1}{K} \sum_{k=1}^{K} \big\| \nabla F_{k,\text{struct}}(P) - \nabla F_{\text{struct}}(P) \big\|_2^2 \\
&\le c^2 \cdot \frac{1}{K} \sum_{k=1}^{K} \|\Delta_k\|_F^2 \\
&= c^2 \cdot \frac{1}{K} \sum_{k=1}^{K} \mathcal{L}_{\text{struct},k}.
\end{aligned}
\tag{44}
$$

which implies Equation (29) (absorbing $c^2$ into $\lesssim$). $\qquad \square$

*Table 8.* Per-domain statistics of the multi-domain datasets. Private is an ultrasound dataset.

| Dataset | Domain | Train | Test | Total |
|---|---|---|---|---|
| FedISIC | MSK | 9,930 | 2,483 | 12,413 |
| | VI-MX | 3,163 | 791 | 3,954 |
| | VI-DL | 2,691 | 672 | 3,363 |
| | RS | 1,807 | 452 | 2,259 |
| | VI-HD | 655 | 164 | 819 |
| | BCN | 351 | 88 | 439 |
| | **Total** | **18,597** | **4,650** | **23,247** |
| FedCamelyon17 | CWZ | 47,548 | 11,888 | 59,436 |
| | RST | 27,923 | 6,981 | 34,904 |
| | UMCU | 68,043 | 17,011 | 85,054 |
| | RUMC | 103,870 | 25,968 | 129,838 |
| | LPON | 117,377 | 29,345 | 146,722 |
| | **Total** | **364,761** | **91,193** | **455,954** |
| Private | RMD | 1,079 | 273 | 1,352 |
| | RMZ | 5,850 | 1,468 | 7,318 |
| | SFY | 5,185 | 1,301 | 6,486 |
| | **Total** | **12,114** | **3,042** | **15,156** |

## B. Implementation

### B.1. Experiment Details

#### B.1.1. DETAILS OF DATASET SETUP

We strictly follow the *one-domain-one-client* protocol to simulate non-IID federated learning settings, where each domain/hospital is treated as a separate client.

- **Datasets:** We evaluate on three medical federated benchmarks spanning optical dermoscopy (FedISIC (Ogier du Terrail et al., 2022a)), histopathology WSI (FedCamelyon17 (Koh et al., 2021)), and ultrasound (Private).

- **Per-domain statistics:** Table 8 reports the train/test split and sample counts for each domain/client.

**Preprocessing.** All images are resized to $224 \times 224$ to match the CLIP input resolution. We apply standard CLIP normalization using mean $(0.481, 0.457, 0.408)$ and standard deviation $(0.268, 0.261, 0.275)$. No aggressive data augmentation is applied during training to strictly evaluate adaptation ability under constrained settings.

#### B.1.2. DETAILS OF IMPLEMENTATION

We implement FedMAP using PyTorch (Paszke et al., 2019). All methods use the pre-trained CLIP ViT-B/16 (Radford et al., 2021) as the backbone, keeping the image and text encoder weights frozen.

- **FedMAP settings:** We insert learnable visual prompts

into all $L = 12$ transformer layers of the image encoder. The prompt length is set to $m_p = 1$ per layer. The structural regularization weight is $\lambda = 10$. EMA momentum is $\mu = 0.9$, and the number of attributes per class is $M = 10$. Main comparison results are averaged over three seeds; unless otherwise specified, ablation and analysis experiments use seed 1.

- **Optimization and FL protocol:** Training is conducted for $T = 50$ communication rounds. In each round, clients perform $E = 1$ local epoch to simulate a resource-constrained medical environment. We use standard FedAvg for server aggregation. For all trainable parameters, we use **SGD** with learning rate $1e-3$, and adopt a **single-step** learning rate scheduler.

- **Hardware:** All experiments are conducted on NVIDIA RTX 3090 GPUs.

**Text-side prompt length for baselines.** For methods involving textual prompt learning, we set the learnable context length to 16 tokens. For LLM knowledge expansion methods, including FedKgCoOP (Yao et al., 2023) and FedMVP (Singha et al., 2025), we use the same expansion protocol for fair comparison.

#### B.1.3. DETAILS OF BASELINE IMPLEMENTATION

We implement and evaluate the following baselines:

- **Zero-shot CLIP**[ICML 2021] (Radford et al., 2021) serves as a non-learning lower bound.

- **PromptFL**[TMC 2023] (Guo et al., 2023b) and **PromptFL+Prox**[PMLR 2020] (Li et al., 2020b) are federated soft prompt tuning baselines.

- **FedKgCoOP** (federated KgCoOP)[CVPR 2023] (Yao et al., 2023) introduces LLM-based knowledge guidance for prompt learning.

- **FedCLIP**[ICLR 2023] (Lu et al., 2023) applies lightweight adapters for federated CLIP adaptation.

- **FedAPT**[AAAI 2024] (Su et al., 2024) employs a class-aware prompt generator to mitigate the class information gap between clients.

- **FedMVP**[ICCV 2025] (Singha et al., 2025) is a multimodal tuning method conditioning prompts on both visual context and textual attribute features.

- **FOCoOP**[ICML 2025] (Liao et al., 2025) is an OOD-aware framework using distribution-level separation and optimal transport under heterogeneous shifts.

*Table 9.* **Domain-wise split-client scalability on FedISIC.** We report Accuracy/Macro-F1 (%) under 2/4/6 clients per original domain.

| Methods | MSK | | VI-MX | | VI-DL | | RS | | VI-HD | | BCN | | Avg | |
|---|---|---|---|---|---|---|---|---|---|---|---|---|---|---|
| | Acc | F1 | Acc | F1 | Acc | F1 | Acc | F1 | Acc | F1 | Acc | F1 | Acc | F1 |
| *One domain split into 2 clients* | | | | | | | | | | | | | | |
| PromptFL [TMC 2023] | 58.32 | 35.7 | 93.55 | 39.8 | 66.37 | 40.5 | 50.44 | 27.9 | 45.73 | 37.9 | 77.27 | 35.1 | 65.28 | 36.2 |
| FedCLIP [ICLR 2023] | 57.51 | 27.5 | 92.79 | 20.7 | 62.65 | 27.1 | 48.89 | 24.7 | 39.02 | 26.8 | 85.23 | 36.9 | 64.35 | 27.3 |
| FedMVP [ICCV 2025] | 55.90 | 26.2 | 93.05 | 21.7 | 60.42 | 23.0 | 43.58 | 19.2 | 38.41 | 22.3 | 73.86 | 29.7 | 60.87 | 23.7 |
| **FedMAP** | **67.58** | **47.3** | **96.33** | **57.7** | **74.70** | **55.4** | **56.42** | **32.0** | **51.83** | **46.4** | **80.68** | **60.4** | **71.26** | **49.9** |
| *One domain split into 4 clients* | | | | | | | | | | | | | | |
| PromptFL [TMC 2023] | 57.83 | 35.5 | 94.06 | 32.7 | 65.03 | 39.4 | 47.35 | 25.7 | 45.12 | 38.3 | 84.09 | 36.0 | 65.58 | 34.6 |
| FedCLIP [ICLR 2023] | 56.42 | 24.5 | 92.79 | 19.4 | 59.08 | 21.1 | 46.46 | 21.1 | 37.80 | 25.8 | 84.09 | 32.3 | 62.78 | 24.0 |
| FedMVP [ICCV 2025] | 55.94 | 26.0 | 93.17 | 28.9 | 60.27 | 22.6 | 40.71 | 16.6 | 40.85 | 23.3 | 78.41 | 30.9 | 61.56 | 24.7 |
| **FedMAP** | **66.94** | **43.1** | **96.71** | **58.0** | **72.62** | **50.3** | **56.42** | **35.6** | **56.71** | **47.8** | **80.68** | **68.0** | **71.68** | **50.5** |
| *One domain split into 6 clients* | | | | | | | | | | | | | | |
| PromptFL [TMC 2023] | 58.80 | 35.0 | 93.43 | 35.7 | 64.29 | 38.5 | 47.12 | 23.9 | 42.68 | 36.8 | 79.55 | 34.4 | 64.31 | 34.1 |
| FedCLIP [ICLR 2023] | 60.25 | 38.1 | 93.17 | 42.4 | 63.54 | 36.1 | 51.33 | 27.4 | 40.24 | 36.1 | 84.09 | 39.0 | 65.44 | 36.5 |
| FedMVP [ICCV 2025] | 55.34 | 23.0 | 93.30 | 19.4 | 59.08 | 20.8 | 40.49 | 16.1 | 39.02 | 20.6 | 78.41 | 27.5 | 60.94 | 21.2 |
| **FedMAP** | **67.82** | **46.1** | **95.58** | **50.6** | **74.11** | **52.3** | **56.64** | **33.2** | **50.00** | **40.6** | **86.36** | **65.6** | **71.75** | **48.1** |

# C. Discussion

## C.1. Effective Rank and Consistency Metrics

In this section, we provide the complete formal definitions of the geometric metrics introduced in the main paper (Section 3 and Section 4.4), including intra- and inter-class effective rank, cross-client Jaccard similarity, and temporal Jaccard stability.

**Effective Rank ($r_{eff}$).** To measure the intrinsic dimensionality of feature geometry (Figure 2), we compute **intra-class** and **inter-class** effective ranks.

**Base definition.** Given a feature matrix $V \in \mathbb{R}^{N \times D}$, let $\Sigma(V) = \frac{1}{N-1}(V - \mathbf{1}\mu^\top)^\top (V - \mathbf{1}\mu^\top)$ be the centered feature covariance, where $\mu = \frac{1}{N}\sum_{n=1}^{N} \mathbf{v}_n$. Let $\lambda_1 \geq \lambda_2 \geq \cdots \geq \lambda_R$ be the eigenvalues of $\Sigma(V)$, with $R = \min\{N-1, D\}$. Define $p_i = \lambda_i / \sum_{j=1}^{R} \lambda_j$. If the eigenvalue sum is zero for a degenerate feature set, we set $r_{eff}(V) = 0$. Otherwise, the effective rank is the exponential of the spectral entropy:

$$r_{eff}(V) = \exp\left( -\sum_{i=1}^{R} p_i \ln p_i \right). \qquad (45)$$

**Intra-class effective rank.** Let $V_c \in \mathbb{R}^{N_c \times D}$ denote the feature matrix of class $c$. We compute $r_{eff}(V_c)$ per class and report the *macro* average:

$$r_{\text{intra}} = \frac{1}{C}\sum_{c=1}^{C} r_{eff}(V_c). \qquad (46)$$

A lower $r_{\text{intra}}$ indicates that within-class features concentrate in a lower-dimensional subspace, reflecting *class-manifold collapse*.

**Inter-class effective rank.** Let $\bar{\mathbf{v}}_c \in \mathbb{R}^D$ be the class mean feature (computed on the same split), and stack class means as $\bar{V} = [\bar{\mathbf{v}}_1; \ldots; \bar{\mathbf{v}}_C] \in \mathbb{R}^{C \times D}$. We define the inter-class effective rank as:

$$r_{\text{inter}} = r_{eff}(\bar{V}), \qquad r_{\text{inter,norm}} = \frac{r_{\text{inter}}}{C-1}. \qquad (47)$$

A lower $r_{\text{inter}}$, or $r_{\text{inter,norm}}$, suggests reduced diversity of class centers and a more entangled inter-class geometry.

**Cross-client Jaccard Similarity.** To quantify Inter-client Topological Misalignment (Figure 3), we measure the pairwise neighborhood consistency across clients. Let $\mathcal{N}_i(x, m)$ denote the set of $m$-nearest neighbors of a sample $x$ retrieved using the visual encoder of client $i$. The Jaccard similarity between client $i$ and client $j$ is defined as:

$$\mathcal{J}_{i,j}(x, m) = \frac{|\mathcal{N}_i(x, m) \cap \mathcal{N}_j(x, m)|}{|\mathcal{N}_i(x, m) \cup \mathcal{N}_j(x, m)|}. \qquad (48)$$

We report the average across all test samples. We set $m = 2$ for Local Consistency and $m = 4$ for Neighborhood Consistency.

**Temporal Jaccard Stability.** For completeness, we restate the temporal Jaccard stability metric introduced in Section 4.4 of the main paper. For each sample $x$, let $\mathcal{N}_2^{(t)}(x)$ be its 2-nearest-neighbor set at round $t$. The neighborhood overlap between consecutive rounds is:

$$\mathcal{J}_{\text{temp}}(x) = \frac{|\mathcal{N}_2^{(t)}(x) \cap \mathcal{N}_2^{(t-1)}(x)|}{|\mathcal{N}_2^{(t)}(x) \cup \mathcal{N}_2^{(t-1)}(x)|}. \qquad (49)$$

A higher $\mathcal{J}_{\text{temp}}$ indicates more stable local structure under federated aggregation. The corresponding analysis is shown in Figure 11.

*Table 10.* **Domain-wise split-client scalability on FedCamelyon17.** We report Accuracy/Macro-F1 (%) under 2/4/6 clients per original domain.

| Methods | CWZ | | RST | | UMCU | | RUMC | | LPON | | Avg | |
|---|---|---|---|---|---|---|---|---|---|---|---|---|
| | Acc | F1 | Acc | F1 | Acc | F1 | Acc | F1 | Acc | F1 | Acc | F1 |
| *One domain split into 2 clients* | | | | | | | | | | | | |
| PromptFL [TMC 2023] | 90.86 | 90.9 | 84.40 | 84.4 | 91.19 | 91.2 | 90.50 | 90.5 | 94.61 | 94.6 | 90.32 | 90.3 |
| FedCLIP [ICLR 2023] | 88.71 | 88.7 | 76.45 | 76.1 | 88.71 | 88.7 | 86.36 | 86.3 | 89.61 | 89.6 | 85.97 | 85.9 |
| FedMVP [ICCV 2025] | 90.74 | 90.7 | 84.83 | 84.8 | 92.39 | 92.4 | 91.95 | 91.9 | 93.71 | 93.7 | 90.72 | 90.7 |
| **FedMAP** | **93.13** | **93.1** | **88.58** | **88.5** | **93.77** | **93.8** | **95.11** | **95.1** | **96.92** | **96.9** | **93.50** | **93.5** |
| *One domain split into 4 clients* | | | | | | | | | | | | |
| PromptFL [TMC 2023] | 90.52 | 90.5 | 84.09 | 84.0 | 90.51 | 90.5 | 89.59 | 89.6 | 93.86 | 93.9 | 89.71 | 89.7 |
| FedCLIP [ICLR 2023] | 85.62 | 85.5 | 71.94 | 71.6 | 86.45 | 86.4 | 84.47 | 84.5 | 87.87 | 87.8 | 83.27 | 83.2 |
| FedMVP [ICCV 2025] | 91.09 | 91.1 | 84.04 | 83.9 | 92.92 | 92.9 | 91.96 | 91.9 | 94.99 | 95.0 | 91.00 | 91.0 |
| **FedMAP** | **92.29** | **92.3** | **87.55** | **87.5** | **93.85** | **93.9** | **93.48** | **93.5** | **95.89** | **95.9** | **92.61** | **92.6** |
| *One domain split into 6 clients* | | | | | | | | | | | | |
| PromptFL [TMC 2023] | 90.88 | 90.9 | 83.18 | 83.1 | 90.86 | 90.9 | 89.90 | 89.9 | 93.55 | 93.5 | 89.68 | 89.7 |
| FedCLIP [ICLR 2023] | 89.19 | 89.2 | 75.86 | 75.5 | 89.82 | 89.8 | 87.44 | 87.4 | 91.74 | 91.7 | 86.81 | 86.7 |
| FedMVP [ICCV 2025] | 90.63 | 90.6 | 83.80 | 83.6 | 91.95 | 91.9 | 91.68 | 91.7 | 94.87 | 94.9 | 90.58 | 90.5 |
| **FedMAP** | **93.55** | **93.5** | **90.37** | **90.4** | **94.56** | **94.6** | **94.64** | **94.6** | **97.04** | **97.0** | **94.03** | **94.0** |

*Table 11.* **Domain-wise split-client scalability on the Private dataset.** We report Accuracy/Macro-F1 (%) under 2/4/6 clients per original domain.

| Methods | RMD | | RMZ | | SFY | | Avg | |
|---|---|---|---|---|---|---|---|---|
| | Acc | F1 | Acc | F1 | Acc | F1 | Acc | F1 |
| *One domain split into 2 clients* | | | | | | | | |
| PromptFL [TMC 2023] | 64.10 | 57.4 | 76.70 | 72.9 | 65.64 | 63.5 | 68.82 | 64.6 |
| FedCLIP [ICLR 2023] | 53.85 | 47.5 | 63.90 | 59.1 | 53.88 | 50.3 | 57.21 | 52.3 |
| FedMVP [ICCV 2025] | 36.26 | 27.7 | 64.99 | 61.0 | 45.66 | 43.7 | 48.97 | 44.1 |
| **FedMAP** | **90.48** | **89.3** | **92.64** | **91.1** | **84.78** | **83.5** | **89.30** | **88.0** |
| *One domain split into 4 clients* | | | | | | | | |
| PromptFL [TMC 2023] | 60.81 | 55.1 | 77.86 | 75.0 | 65.80 | 63.6 | 68.15 | 64.6 |
| FedCLIP [ICLR 2023] | 47.25 | 40.5 | 53.95 | 47.5 | 46.12 | 41.5 | 49.11 | 43.2 |
| FedMVP [ICCV 2025] | 45.79 | 40.0 | 56.47 | 52.8 | 44.97 | 41.7 | 49.07 | 44.8 |
| **FedMAP** | **82.05** | **76.9** | **91.01** | **87.7** | **82.63** | **80.7** | **85.23** | **81.8** |
| *One domain split into 6 clients* | | | | | | | | |
| PromptFL [TMC 2023] | 58.97 | 53.1 | 76.43 | 71.4 | 62.87 | 59.7 | 66.09 | 61.4 |
| FedCLIP [ICLR 2023] | 40.29 | 33.8 | 45.10 | 38.6 | 41.05 | 35.6 | 42.14 | 36.0 |
| FedMVP [ICCV 2025] | 34.80 | 27.4 | 56.81 | 49.7 | 39.97 | 36.1 | 43.86 | 37.7 |
| **FedMAP** | **80.59** | **76.0** | **89.37** | **84.4** | **80.17** | **76.9** | **83.38** | **79.1** |

## C.2. Dataset-wise Discussion

Table 1–Table 3 show that FedMAP achieves the best **Accuracy** and **Macro-F1** across dermoscopy (FedISIC (Ogier du Terrail et al., 2022a)), WSI histopathology (FedCamelyon17 (Koh et al., 2021)), and ultrasound (Private) while using lightweight learnable parameters. The gains are especially informative against FedKgCoOP (Yao et al., 2023) and FedMVP (Singha et al., 2025): knowledge-guided prompts may introduce non-visual semantics, whereas multimodal tuning can be sensitive to inter-client gradient conflicts. FedMAP instead uses visually grounded anchors plus relation alignment, yielding a more stable cross-client geometry under heterogeneous medical shifts.

## C.3. Supplementary Details for Ablation Studies and In-depth Analysis

**Anchor construction variants.** For Table 4, **CLS-only** uses only the class-name embedding; **Template** uses "a photo of a {cls}"; **Attribute-only** uses all LLM-generated attributes without the class name; **Single** samples one "{cls}:{attribute}" anchor per class; and **MSA** ensembles the full "{cls}:{attribute}" codebook.

**Alignment loss details.** For Table 5, the KL variant treats the softmax-normalized text relation matrix as the target and minimizes KL divergence from the visual-side softmax-normalized similarity, using $\tau_{KL} = 1.0$.

**t-SNE setup.** All t-SNE/Jaccard variants use frozen CLIP ViT-B/16: **Text Tuning** (**Static**) tunes only text prompts, **FedCLIP** adapts lightweight visual modules, **Text+Visual Tuning** jointly tunes text and visual modules, and **FedMAP** (**Ours**) applies MSA and TSA.

# D. Additional Experiments

## D.1. Split-client Scalability

To test client-count sensitivity, we split each original domain into 2, 4, and 6 clients. As shown in Tables 9 to 11, FedMAP remains stable across these fine-grained partitions, indicating that semantic anchoring and topology alignment scale beyond the standard hospital/domain-level protocol.

