# OpenReview forum: "Rethinking Federated Prompt Learning for Medical Images: From Textual Tuning to Visual Manifold Anchoring"
_ICML.cc/2026/Conference — ICML 2026 regular_

### Official Review · Reviewer_nEWt · 2026-03-04

**Soundness:** 2
**Presentation:** 3
**Significance:** 2
**Originality:** 3
**Overall Recommendation:** 4
**Confidence:** 4

**Summary:**

This paper proposes FedMAP, a framework for federated prompt learning in medical imaging. The authors identify two geometric issues when using frozen CLIP backbones in federated settings: intra-client manifold collapse and inter-client topological misalignment. To address these, they construct a shared semantic codebook using LLM-generated class attributes encoded by a frozen text encoder, which serves as a client-invariant reference. During federated training, only lightweight visual prompts are optimized, guided by (i) an anchor-based classification loss (MSA) and (ii) a structural alignment loss (TSA) that aligns class similarity structures across clients. Experiments on three medical FL benchmarks show consistent improvements over prior prompt-based federated methods, along with improved convergence and communication efficiency.

**Compliance With Llm Reviewing Policy:**

Affirmed.

**Key Questions For Authors:**

1. Could you explicitly report the number of clients, number of communication rounds, local training epochs, and the degree of data heterogeneity for each dataset? Additionally, how are samples distributed across clients?
2.  Have you evaluated FedMAP and baselines across multiple random seeds? If so, could you report mean ± standard deviation and statistical significance tests?
3.  Have you evaluated FedMAP under settings with more clients or higher heterogeneity (e.g., synthetic splits or client subsampling)? If not, can you comment on expected scalability in such scenarios?
4.  The theory suggests structural alignment reduces client drift. Do you have empirical measurements of inter-client drift (e.g., representation divergence across rounds) that directly validate this claim?
5.  How sensitive is performance to the quality or wording of LLM-generated attributes? For example, what happens if attributes are noisy, simplified, or manually defined?
6.  The paper leverages both image and text encoders (via CLIP) and introduces LLM-derived semantic anchors. However, the model does not perform joint cross-modal training or generative text–image interaction, but rather uses text embeddings as fixed semantic priors for visual alignment. Could the authors clarify how they position FedMAP relative to multimodal learning? In particular, do they consider this a multimodal learning framework, or a visually grounded model guided by frozen textual priors?

**Limitations:**

The discussion of limitations and societal impact is relatively brief and could be strengthened. In particular, the paper would benefit from addressing the scalability of FedMAP to larger federated systems with many clients and highly heterogeneous data distributions, which is important for real-world deployment (e.g., distributed healthcare networks). The method’s reliance on LLM-generated semantic anchors also raises potential concerns regarding sensitivity to prompt wording, bias in LLM outputs, and possible mismatch between LLM priors and medical reality. Additionally, the robustness of the approach across different backbone models, data modalities, and client distributions is not clearly explored. Finally, while federated learning improves privacy, the use of external LLM-derived semantic anchors may introduce subtle privacy or information leakage considerations when class descriptions are domain-specific. A brief dedicated limitations discussion covering these aspects would strengthen the completeness of the paper.

**Strengths And Weaknesses:**

Soundness: The paper is technically coherent and well-motivated. The geometric framing of intra-client manifold collapse and inter-client topological misalignment is clearly linked to the proposed objectives (MSA and TSA). The use of an LLM-derived semantic codebook as a shared reference across clients is internally consistent, and the structural alignment loss directly targets cross-client relational consistency. The experimental evaluation includes multiple benchmarks, convergence analysis, communication cost comparisons, and ablations, and the theoretical section provides a drift-related analysis under federated optimization. However, the federated protocol lacks sufficient detail: the number of clients, number of communication rounds, heterogeneity level, and data distribution per client are not clearly specified, limiting reproducibility and assessment of realism. Scalability to larger or more heterogeneous federated settings is not evaluated. Results are reported without variance across random seeds or statistical testing. While the theory is structured, empirical validation of drift reduction could be more direct. Performance gains over strong baselines are consistent but sometimes moderate, and stronger stress testing would increase confidence.

Presentation: The paper is clearly written and logically structured, with a coherent geometric narrative connecting motivation, method, and theory. The two-stage pipeline is easy to follow, and figures effectively illustrate the intuition behind semantic anchoring and structural alignment. Some claims (e.g., describing the LLM as a “structural blueprint”) slightly overstate the mechanism, which functions more as structured semantic regularization. The structural alignment process could be more explicitly illustrated, and reproducibility details (e.g., baseline tuning procedures) could be clarified.

Significance: The work addresses a relevant problem: adapting frozen multimodal foundation models under federated constraints in a communication-efficient manner. The idea of injecting LLM-derived semantic anchors as shared cross-client priors is practically appealing and may inspire further research on structure-aware federated learning. However, the empirical gains, while consistent, are incremental rather than transformative, and scalability beyond the tested setting is not demonstrated. The broader impact on general federated learning remains to be established.

Originality: The paper provides a meaningful geometric reframing of federated prompt learning and introduces a creative integration of semantic anchoring with relational structural alignment. The use of an LLM-derived codebook as a client-invariant synchronization signal is conceptually interesting. Nevertheless, individual components (prompt tuning, semantic priors, prototype or similarity alignment) build on existing ideas. The contribution lies more in structured integration and reformulation than in a fundamentally new learning paradigm. Originality is therefore moderate but well-justified.

---

> ### Author Rebuttal · Authors · 2026-03-30
>
> **We sincerely appreciate the reviewer's time and effort.**
>
> # Q1. Clarification of the Federated Protocol
>
> **R1.** Sorry that the protocol description was not clear enough. FedISIC, FedCamelyon17, and Private use 6, 5, and 3 clients, with 50 rounds and 1 local epoch per round. Heterogeneity comes from site/hospital/domain shift, with one site/domain per client.
>
> **Table 1. Main federated protocol.**
>
> | Dataset | Clients | Rounds | Local Epochs | Sample Distribution |
> | --- | ---: | ---: | ---: | --- |
> | FedISIC | 6 | 50 | 1 | One site per client |
> | FedCamelyon17 | 5 | 50 | 1 | One hospital per client |
> | Private | 3 | 50 | 1 | One site per client |
>
> # Q2. Multi-Seed Evaluation and Statistical Testing
>
> **R2.** Thank you for asking for multi-seed results and statistical testing. Across three seeds, FedMAP remains best on three benchmarks and is significant against baselines.
>
> **Table 2. Multi-seed overall averages (mean ± std).**
>
> | Method | FedISIC Acc | FedISIC F1 | FedCam Acc | FedCam F1 | Private Acc | Private F1 |
> | --- | ---: | ---: | ---: | ---: | ---: | ---: |
> | PromptFL | 65.11±0.48 | 39.90±2.82 | 90.26±0.35 | 90.22±0.35 | 68.01±0.87 | 63.90±1.46 |
> | FedCLIP | 66.13±0.38 | 33.38±1.28 | 88.70±0.32 | 88.67±0.33 | 61.58±3.99 | 57.48±4.78 |
> | FedMVP | 61.98±0.51 | 26.04±2.26 | 88.81±3.10 | 88.75±3.09 | 51.72±2.67 | 46.86±3.28 |
> | FedMAP (Ours) | **73.29±1.29** | **59.47±1.98** | **93.15±0.87** | **92.99±0.81** | **89.60±1.33** | **88.04±1.83** |
>
> **Table 3. Statistical summary of FedMAP vs. trainable baselines (significant at $p<0.05$).**
>
> | Baseline | FedISIC p(Acc/F1) | Sig. | FedCam p(Acc/F1) | Sig. | Private p(Acc/F1) | Sig. |
> | --- | --- | ---: | --- | ---: | --- | ---: |
> | PromptFL | 0.0051 / 0.0095 | √ | 0.0254 / 0.0251 | √ | 0.0024 / 3.73e-04 | √ |
> | FedCLIP | 0.0037 / 1.37e-04 | √ | 0.0026 / 0.0022 | √ | 0.0049 / 0.0062 | √ |
> | FedMVP | 0.0027 / 8.29e-04 | √ | 0.0488 / 0.0400 | √ | 7.22e-04 / 0.0012 | √ |
>
> # Q3. Evaluation under Larger Client Counts
>
> **R3.** Thank you for asking about larger client counts. We split each original domain into 2/4/6 clients and reevaluate `PromptFL`, `FedCLIP`, `FedMVP`, and `FedMAP`. Under these more heterogeneous split-client settings, FedMAP keeps the highest accuracy on all three datasets.
>
> **Table 4. More-clients evaluation with 2 clients per domain (Avg Acc).**
>
> | Method | FedCamelyon17 | FedISIC | Private |
> | --- | ---: | ---: | ---: |
> | PromptFL | 90.32 | 65.28 | 68.82 |
> | FedCLIP | 85.97 | 64.35 | 57.21 |
> | FedMVP | 90.72 | 60.87 | 48.97 |
> | FedMAP (Ours) | **93.50** | **71.26** | **89.30** |
>
> **Table 5. More-clients evaluation with 4 clients per domain (Avg Acc).**
>
> | Method | FedCamelyon17 | FedISIC | Private |
> | --- | ---: | ---: | ---: |
> | PromptFL | 89.71 | 65.58 | 68.15 |
> | FedCLIP | 83.27 | 62.78 | 49.11 |
> | FedMVP | 91.00 | 61.56 | 49.07 |
> | FedMAP (Ours) | **92.61** | **71.68** | **85.23** |
>
> **Table 6. More-clients evaluation with 6 clients per domain (Avg Acc).**
>
> | Method | FedCamelyon17 | FedISIC | Private |
> | --- | ---: | ---: | ---: |
> | PromptFL | 89.68 | 64.31 | 66.09 |
> | FedCLIP | 86.81 | 65.44 | 42.14 |
> | FedMVP | 90.58 | 60.94 | 43.86 |
> | FedMAP (Ours) | **94.03** | **71.75** | **83.38** |
>
> # Q4. Validation of Drift Reduction
>
> **R4.** Thank you for asking for a drift metric. We quantify inter-client consistency by Jaccard overlap and report the round-wise change below. FedMAP improves local consistency much more strongly than the baselines, supporting reduced inter-client drift.
>
> **Table 7. Local consistency (Jaccard) across communication rounds.**
>
> | Method | R0 | R10 | R20 | R30 | R40 | R50 |
> | --- | ---: | ---: | ---: | ---: | ---: | ---: |
> | Text Tuning | 0.22 | 0.22 | 0.22 | 0.22 | 0.22 | 0.22 |
> | FedCLIP | 0.25 | 0.30 | 0.38 | 0.39 | 0.35 | 0.36 |
> | Text + Visual | 0.33 | 0.42 | 0.46 | 0.47 | 0.46 | 0.48 |
> | FedMAP (Ours) | **0.30** | **0.55** | **0.52** | **0.50** | **0.52** | **0.57** |
>
> # Q5. Sensitivity to LLM-Derived Semantic Anchors
>
> **R5.** Thank you for raising the anchor-sensitivity issue. We answer this with an anchor-construction ablation. FedMAP is strongest with the full MSA codebook ensemble, while weaker or noisier anchors reduce performance in a consistent way.
>
> **Table 8. Anchor / noisy MSA summary (Avg Acc).**
>
> | Setting | Avg |
> | --- | ---: |
> | Random anchors | 62.67 |
> | Class name only | 84.12 |
> | Attribute-only | 84.85 |
> | Noisy MSA ($\sigma=0.01$) | 83.78 |
> | Noisy MSA ($\sigma=0.05$) | 79.78 |
> | Noisy MSA ($\sigma=0.10$) | 53.84 |
> | Single-anchor MSA | 83.50 |
> | **MSA codebook ensemble (ours)** | **85.74** |
>
> The drop under stronger noise and weaker anchors shows that FedMAP benefits from semantic quality without depending on a fixed wording.
>
> # Q6. FedMAP in the Multimodal Setting
>
> **R6.** Thank you for pointing this out. FedMAP is better described as a **vision-centric federated adaptation method guided by frozen textual priors**, not a full joint multimodal framework. We will clarify this in revision.

---

### Official Review · Reviewer_8JvS · 2026-03-10

**Soundness:** 3
**Presentation:** 3
**Significance:** 3
**Originality:** 3
**Overall Recommendation:** 4
**Confidence:** 4

**Summary:**

In this paper, two geometric failure modes of existing federated prompt learning  methods are identified when applied to medical imaging: (1) intra-client medical manifold collapse, that is, the high degree of morphological similarity between medical classes compresses features to a low-rank subspace; (2) inter-client topological misalignment, where heterogeneous acquisition protocols across hospitals produce inconsistent neighborhood structures. To address these issues, the authors propose FedMAP, a method that shifts from text tuning to visual manifold anchoring. FedMAP builds a semantic codebook by querying a LLM for fine-grained visual attributes for each category, and encodes it through a frozen CLIP text encoder to form a client-invariant anchor. Two mechanisms are introduced: manifold semantic anchoring, which uses these anchors as classification references to extend effective subspaces; Topology Structral Alignment which aligns the local visual relationship matrix with the text-derived relationship matrix. Experiments on FedISIC, FedCamelyon17, and a private ultrasound dataset have shown consistent performance improvements over the strong baseline approach.

**Compliance With Llm Reviewing Policy:**

Affirmed.

**Final Justification:**

My concerns have been adequately addressed, and my final recommendation is weak accept.

**Key Questions For Authors:**

Q1. How does FedMAP scale to having more clients (e.g., 20-50) and clients distributed across shared domain names? The current “one domain, one client” protocol for 3-6 clients is far from realistic for a hospital consortium.

Q2. In the prototype Q, updates are performed via EMA (Equation 6) each round. Are they re-initialized from the global prompt at every communication round, or do they persist across each client’s rounds? How sensitive is the method to the EMA momentum μ=0.9?

Q3. Can the TSA framework be extended to tasks without discrete class structures, such as regression or dense prediction (segmentation)?

**Limitations:**

yes

**Strengths And Weaknesses:**

## Strengths

1) The paper conducts a rigorous geometric diagnosis of the reasons for failure in medical FL Chinese text fine-tuning. Convincing quantitative evidence is provided through effective ranking (see Figure 2) and Jaccard neighborhood consistency (see Figure 3). The experimental setup covers three different medical imaging modalities (ablation studies (Tables 4-5, Figures 5-6, 8) are detailed).

2) The paper is well written, with a clear narrative arc from problem identification to solution. Figure 1 effectively visualizes the motivation and method. Figure 4 clearly shows the two-phase pipeline.

3) This issue is very important in practice: federated learning in medical imaging is developing, and understanding why existing online tuning methods fail geometrically is very valuable. Communication efficiency is significant: FedMAP requires only 0.07 MB per round (see Table 8), which is three times lower than FedMVP (0.23 MB), while achieving better performance.

4) The geometric perspective on FPL's failure is novel and well-supported. Using an LLM-derived semantic codebook as a frozen anchor combined with topologically aware relational distillation is a creative design. While previous work included individual components (such as LLM attribute generation, prototype-based functions, and relational knowledge distillation), integrating them into a coherent framework anchored to the visual manifold is new.

## Weaknesses

1) The federated setting is unrealistically small. This is my biggest concern. Each client maps to exactly one hospital/domain, giving K=6, K=5, or K=3 clients. Real hospital consortia have tens to hundreds of institutions, often with overlapping distributions. The one-domain-one-client setup makes the topology alignment story clean but potentially misleading.

2) Only classification is tested. Medical FL regularly involves segmentation and detection. The entire framework, class-wise prototypes, C x C relation matrices, assumes discrete classes. Extending to dense prediction is non-obvious, and the paper barely describe this gap.

3) The relationship between MSA and prototype-based FL methods like FedProto and FedFA deserves explicit discussion. The anchoring mechanism is conceptually similar, with the difference being frozen text anchors vs. learned prototypes.

---

> ### Author Rebuttal · Authors · 2026-03-30
>
> **We sincerely appreciate the reviewer's time and effort.**
>
> # Q1 & W1. Scalability to More Clients under Shared-Domain Federation
>
> **R1.** Thank you for asking about larger-client settings. We extend the shared-domain evaluation to `2/4/6` clients per domain on the three benchmarks. Under these more heterogeneous split-client settings, FedMAP still maintains high accuracy and consistently outperforms the baselines.
>
> **Table 1. More-clients evaluation with 2 clients per domain.**
>
> | Method | FedCamelyon17 | FedISIC | Private |
> | --- | ---: | ---: | ---: |
> | PromptFL | 90.32 | 65.28 | 68.82 |
> | FedCLIP | 85.97 | 64.35 | 57.21 |
> | FedMVP | 90.72 | 60.87 | 48.97 |
> | FedMAP (Ours) | **93.50** | **71.26** | **89.30** |
>
> **Table 2. More-clients evaluation with 4 clients per domain.**
>
> | Method | FedCamelyon17 | FedISIC | Private |
> | --- | ---: | ---: | ---: |
> | PromptFL | 89.71 | 65.58 | 68.15 |
> | FedCLIP | 83.27 | 62.78 | 49.11 |
> | FedMVP | 91.00 | 61.56 | 49.07 |
> | FedMAP (Ours) | **92.61** | **71.68** | **85.23** |
>
> **Table 3. More-clients evaluation with 6 clients per domain.**
>
> | Method | FedCamelyon17 | FedISIC | Private |
> | --- | ---: | ---: | ---: |
> | PromptFL | 89.68 | 64.31 | 66.09 |
> | FedCLIP | 86.81 | 65.44 | 42.14 |
> | FedMVP | 90.58 | 60.94 | 43.86 |
> | FedMAP (Ours) | **94.03** | **71.75** | **83.38** |
>
> These results show that the advantage of FedMAP remains clear as client count increases.
> The gap is especially clear on the more heterogeneous Private benchmark, indicating that FedMAP remains robust even when each original domain is split into finer-grained client partitions.
>
> # Q2. Clarification about Prototype-Bank Persistence and EMA Sensitivity
>
> **R2.** Sorry that the prototype-bank update was not clearly described. The prototype bank $Q$ is not re-initialized at each communication round. It is kept as a persistent buffer and updated by EMA during local training.
>
> **Table 4. Sensitivity analysis of the EMA momentum $\mu$ on three benchmarks.**
>
> | Dataset | 0.1 | 0.2 | 0.3 | 0.4 | 0.5 | 0.6 | 0.7 | 0.8 | 0.9 |
> | --- | ---: | ---: | ---: | ---: | ---: | ---: | ---: | ---: | ---: |
> | FedCamelyon17 | 91.54 | 92.58 | 91.57 | 92.62 | 91.55 | 91.57 | 91.63 | 91.49 | 93.64 |
> | FedISIC | 72.77 | 72.62 | 72.67 | 72.72 | 72.79 | 72.63 | 72.31 | 72.52 |  73.23 |
> | Private | 89.28 | 89.92 | 88.49 | 88.64 | 89.65 | 89.47 | 89.95 | 89.69 |90.36  |
>
> FedMAP is robust to $\mu$ over a broad range. We use $\mu=0.9$ because it gives the strongest and most consistent results across the three benchmarks.
>
> # Q3 & W2. Further Discussion of Extending TSA beyond Classification Tasks
>
> **R3.** Thank you for asking about settings beyond classification. TSA aligns relations among semantic units, so it is not restricted to classification. In the current paper, those units are class prototypes $ \lbrace q_c \rbrace_{c=1}^C$, and TSA aligns
>
>  $$
> R^{v}\_{ij} = s(q_i^{v}, q_j^{v}), \quad R^{t}\_{ij} = s(q_i^{t}, q_j^{t}), \quad \mathcal{L}_{\text{TSA}} = d(R^v, R^t).
>  $$
>
> For dense prediction, class prototypes can be replaced by region-, pixel-, or query-level semantic units. For regression, let $  \lbrace u_m \rbrace_{m=1}^M $ be ordered target anchors. TSA can align
>
>  $$
> R^{y}\_{ij}=s(u_i,u_j), \quad R^{v}\_{ij}=s(z_i,z_j), \quad \mathcal{L}_{\text{reg-TSA}}=d(R^v,R^y),
>  $$
>
> where $z_i$ denotes the visual embedding associated with anchor $u_i$. The semantic units change across tasks, but the relation-alignment principle remains the same.
>
> # W3. Further Discussion on the Relation Between MSA and Prototype-Based FL
>
> **R4.** Thank you for pointing out the connection to prototype-based FL. MSA is related to prototype-based FL because both introduce a class-level reference, but the role of that reference is different.
>
> In prototype-based FL such as FedProto/FedFA, the class prototype is **learned from client visual features**. A typical form is
>
>  $$
> p_c^{(k)}=\frac{1}{|D_c^{(k)}|}\sum_{(x,y)\in D_c^{(k)},\,y=c} f_k(x),
> \qquad
> \bar p_c=\text{Agg}\big(\{p_c^{(k)}\}_{k=1}^K\big),
>  $$
>
> The regularization then pulls local features toward these learned visual prototypes:
>
>  $$
> \mathcal{L}_{\text{proto}}=\sum_c \|p_c^{(k)}-\bar p_c\|_2^2.
>  $$
>
> In contrast, MSA does **not** learn or synchronize class prototypes across clients. Instead, it constructs a fixed semantic anchor set from text:
>
>  $$
> a_c = E_{\text{text}}(t_c),
>  $$
>
> where $t_c$ is the semantic description/codebook of class $c$, and $E_{\text{text}}(\cdot)$ is the frozen text encoder. The visual feature $z=f_k(x)$ is then classified against these fixed anchors by
>
>  $$
> \ell_c = \frac{z^\top a_c}{\tau}, \quad  \mathcal{L}\_{\text{MSA}}=\mathrm{CE}(\{\ell_c\}_{c=1}^C, y).
>  $$
>
> FedProto/FedFA use **client-dependent learned visual prototypes** and reduce cross-client discrepancy by prototype aggregation, whereas MSA uses a **client-invariant fixed semantic codebook**.
> This difference is precisely why MSA is complementary to prototype-based FL but not equivalent to it.

---

> > ### Author Rebuttal · Reviewer_8JvS · 2026-04-02
> >
> > My concerns have been adequately addressed.

---

> > > ### Author Response · Authors · 2026-04-05
> > >
> > > We sincerely thank the reviewer for the positive feedback and for carefully considering our rebuttal.
> > >
> > > We are very pleased that our responses were able to adequately address the concerns raised. The reviewer’s comments have been very valuable in improving the clarity and quality of the paper.
> > >
> > > Thank you again for your valuable feedback and consideration!

---

### Official Review · Reviewer_Pqpy · 2026-03-18

**Soundness:** 3
**Presentation:** 3
**Significance:** 3
**Originality:** 3
**Overall Recommendation:** 4
**Confidence:** 4

**Summary:**

This paper identifies geometric limitations of federated prompt learning in medical vision-language models, including intra-client manifold collapse and inter-client topological misalignment. To address these issues, it proposes FedMAP, a visual manifold anchoring framework using an LLM-derived codebook for semantic and structural alignment. In summary, the authors propose a paradigm to tackle the fundamental limitations of textual prompt tuning in medical federated learning. Generally, the motivation is well explained, and the proposed solution is easy to follow and seemingly effective.

**Compliance With Llm Reviewing Policy:**

Affirmed.

**Ethical Review Concerns:**

No ethics required

**Final Justification:**

Thanks for the rebuttal. The author has addressed my main concerns, and I will raise my score. In Final Justification,  I have no further concerns.

**Key Questions For Authors:**

Questions:
1.The ablation in the Tables shows joint MSA+TSA gains, but the individual and interaction effects are summarized rather than fully tabulated. Could the authors add a complete 2×2 ablation (MSA on/off × TSA on/off) to show the performance gain brought by each component?
2.The private ultrasound dataset is critical for claiming real-world applicability, yet its general description, such as class distribution, label noise level, and scanner/protocol differences are not disclosed. Could the authors at least release anonymized class-distribution statistics and a short description of acquisition heterogeneity?
3.Figure 10 shows effective-rank dynamics only on the three medical datasets. Could the authors include the same plot for one strong natural FL baseline (e.g., CIFAR-100 FL suggested or PACS already used) to comparatively quantify how much more severe the collapse is in the medical setting?
4.Again, the method shares notable technical similarities with FedMVP (LLM semantic enrichment + visual-only prompting) and FedDecorr (also in F-norm but used as regularization). Could the authors clarify the precise technical novelty of MSA+TSA over these prior works with more in-depth details and discussions.

**Limitations:**

See weakness.

**Strengths And Weaknesses:**

Strength:
1.The motivation of this paper is clear.
2.The paper is well written, and the technical details are well explained and easily understood.
3.The reported benchmark can support the effectiveness of the proposed method and performance improvement claimed by the author.
Weakness:
1.Lacking validating Manifold Collapse in FL nature image dataset: In my personal view, the main contributions and insight the paper may bring to the community will be its motivation, namely Intra-client: Medical Manifold Collapse and Inter-client: Medical Topological Misalignment. As I put in the Strength part, it is well written. However, as the scope of the paper is within FL, it could be better to evaluate the medical manifold collapse in more widely-used FL dataset beyond PACS (although also a common dataset, but more tailored for domain generalization).
2.The contribution part seems ambiguous in the current version, and thus makes the overall technical contribution seemingly overclaimed after full check because the solution is unexpectedly simple. The main technical contributions seem to propose two training strategies, or more specifically two loss functions, to tackle the challenges. In the introduction Section, general description of the solutions is somehow absent.
3.In Sec 3.3.2 and total loss of Eqn. (9), L_CE seems sharing the very similar formulation of the CLIP cross-entropy loss of FedMVP in Eqn (3)-(4). The author can make more explanations here about the differences, or the reasons for making such choices.
4.TSA uses F-norm between the relation matrix of texts and images to align the intrinsic connectivity, and make the ablation study with KL divergence. Its motivation is clear and the solution is efficient. However, I think the other type of distance induced by matrix norm, could be ablation candidates beyond KL divergence. If I were not wrong, KL-divergence is a less common candidate for such purpose. As the so-called relation matrix seems very similar to the general correlation matrix, other distance used to regularize the correlation matrix, like “1 – cosine similarity” in the Eqn. (5) of FedMVP and widely-used Bures–Wasserstein distance in ML community, are encouraged to show in ablation study to fully support the rationality of the choices.  It would be great if there are relevant experimental instructions and explanations.
5.Limitations of the work can be presented for better understanding and future research.

---

> ### Author Rebuttal · Authors · 2026-03-30
>
> **We sincerely appreciate the reviewer's time and effort.**
>
> # Q1. Clarification through a Full 2×2 Ablation of MSA and TSA
>
> **R1.** Thank you for requesting the full ablation. We provide the complete 2×2 results below on both FedISIC and the private dataset. Both MSA and TSA improve over the baseline, and their combination is best.
>
> **Table 1. Full 2×2 ablation of MSA and TSA.**
>
> | MSA | TSA | FedISIC | Private |
> | --- | --- | ---: | ---: |
> | ✗ | ✗ | 71.55 | 88.00 |
> | ✓ | ✗ | 72.62 | 89.70 |
> | ✗ | ✓ | 72.80 | 88.70 |
> | ✓ | ✓ | **73.23** | **90.36** |
>
> These results show that MSA and TSA are complementary rather than redundant, since each individual component improves over the baseline while their combination performs best on both datasets.
>
> # Q2. Clarification about the Private Ultrasound Benchmark and Its Heterogeneity
>
> **R2.** Sorry that the dataset description was not clear enough. We provide anonymized site-level statistics below. The private benchmark contains 13 fetal ultrasound standard-plane classes from 3 sites. Heterogeneity comes from cross-site differences in acquisition conditions, workflow, and clinical environment.
>
> **Table 2. Anonymized site-level statistics of the private benchmark.**
>
> | Dataset | Site | Train | Test | Total |
> | --- | --- | ---: | ---: | ---: |
> | Private | RMD | 1,079 | 273 | 1,352 |
> | Private | RMS | 5,850 | 1,468 | 7,318 |
> | Private | SFY | 5,185 | 1,301 | 6,486 |
> | Private | **Total** | **12,114** | **3,042** | **15,156** |
>
> The benchmark is therefore clinically meaningful and naturally heterogeneous at the client level.
>
> # Q3 & W1. Further Natural-Image Validation of the Geometric Collapse Claim
>
> **R3.** Thank you for asking for validation beyond medical data. PACS provides the requested natural-image **domain-skew** reference. We report both round-wise geometric indicators below.
>
> **Table 3a. PACS intra-class effective rank.**
>
> | Method | R1 | R10 | R20 | R30 | R40 | R50 |
> | --- | ---: | ---: | ---: | ---: | ---: | ---: |
> | FedCLIP | 40.98 | 39.03 | 38.40 | 37.85 | 37.68 | 37.77 |
> | VPT-LPT | 40.51 | 37.03 | 35.14 | 34.63 | 33.30 | 33.54 |
> | FedMAP (Ours) | **40.38** | **39.21** | **39.37** | **40.16** | **41.74** | **41.69** |
>
> **Table 3b. PACS inter-class effective rank.**
>
> | Method | R1 | R10 | R20 | R30 | R40 | R50 |
> | --- | ---: | ---: | ---: | ---: | ---: | ---: |
> | FedCLIP | 0.835 | 0.830 | 0.829 | 0.831 | 0.832 | 0.837 |
> | VPT-LPT | 0.833 | 0.838 | 0.854 | 0.856 | 0.859 | 0.861 |
> | FedMAP (Ours) | **0.835** | **0.837** | **0.851** | **0.858** | **0.862** | **0.869** |
>
> These two indicators show the same pattern on PACS, while the effect remains more pronounced on the medical datasets.
> This supports our claim that the geometric-collapse phenomenon is not limited to medical federation, even though it is more visible there.
>
> # Q4 & W2 & W3 & W4. Further Discussion of the Precise Technical Novelty Relative to Existing Methods
>
> **R4.** Thank you for asking for a sharper discussion of technical novelty. FedMAP uses a different formulation of federated prompt learning. It constructs a frozen semantic anchor matrix
>
>  $$
> a_c=\mathrm{norm}(\left(\frac{1}{M}\sum_m E_T(\phi(c,s_{c,m}))\right)), \quad A=[a_1,\ldots,a_C],
>  $$
>
> uses it for anchor-based classification
>
>  $$
> p(y=c\mid x)=\frac{\exp(\mathrm{sim}(v,a_c)/\tau)}{\sum_j \exp(\mathrm{sim}(v,a_j)/\tau)},
>  $$
>
> and aligns visual and semantic relations by
>
>  $$
> G_{\text{text}}=AA^\top,\quad G_{\text{vis}}=QQ^\top,\quad
> L_{\text{struct}}=\|G_{\text{vis}}-G_{\text{text}}\|_F^2.
>  $$
>
> The key novelty relative to prior work lies in what is optimized. FedMVP learns image-conditioned prompts via
>
>  $$
> P(A',E)=\mathrm{FFN}(\mathrm{CrossAttention}(Q_E,K_{A'},V_{A'})),\qquad
> L=L_{ce}+\alpha L_{con},
>  $$
>
> so its CE term supervises a **dynamic prompt generator**. In contrast, FedMAP classifies against a **fixed semantic codebook**. FedDecorr regularizes feature-dimension correlation by
>
> $$
> L_{\text{FD}}=\frac{1}{d}\|K\|_F^2.
> $$
>
> Thus FedMAP is based on **fixed semantic anchoring + class-topology alignment**, not dynamic prompt generation or generic decorrelation. The TSA distance is further justified empirically by extending the ablation from MSE/KL to cosine-style matching and Bures--Wasserstein distance. In the revision, we will also explicitly cite and discuss FedMVP, FedDecorr, and the most relevant related methods in the main text.
>
> **Table 4. Ablation of TSA relation distances.**
>
> | Relation Distance | FedISIC | Private |
> | --- | ---: | ---: |
> | KL divergence | 70.99 | 88.89 |
> | 1 - cosine similarity | 72.79 | 89.72 |
> | Bures--Wasserstein | 72.17 | 89.43 |
> | MSE (ours) | **73.06** | **90.36** |
>
> These results support both the distinction and the MSE choice.
>
> # W5. Further Discussion of Limitations and Scope
>
> **R5.** Thank you for this suggestion. The main limitation is that the current paper validates FedMAP only on **classification**. We will add a limitations paragraph to make this scope explicit.

---

> > ### Author Rebuttal · Reviewer_Pqpy · 2026-04-04
> >
> > Thanks for the rebuttal. The author has addressed my main concerns, and I will raise my score.

---

> > > ### Author Response · Authors · 2026-04-05
> > >
> > > Thank you very much for your positive feedback and for taking the time to carefully review our rebuttal.
> > >
> > > We sincerely appreciate your thoughtful comments throughout the review process. We are very pleased that our rebuttal was able to address your main concerns, and we also truly appreciate your willingness to reconsider and raise the score.
> > >
> > > Thank you again for your valuable feedback and support！

---

### Decision · Program_Chairs · 2026-04-30

**Decision:**

Accept (regular)

**Comment:**

In this paper the authors presented a method about federated prompt learning for medical images. The paper was reviewed by three expert reviewers, followed by a rebuttal and discussion between the reviewers and authors. The paper received an overall positive rating with three Weak Accept.

The reviewers agree on the motivation and practical value of the studied problem, the effectiveness of the proposed method and the validation, the presentation and the writing quality. Clear contributions could be made to the community, as appreciated by the reviewers.
There were some concerns regarding the weaknesses of the paper, including the missing details and experimental analysis, technical novelty, some overclaims, and the significance of the performance. After the rebuttal and the authors-reviewers discussion phase, most concerns were well addressed, as acknowledged by the reviewers.
Overall, the paper is technically sound, well-written, and would be a good contribution to at least some fraction of the ICML community.

As a result, the AC is happy to recommend acceptance of this paper, but the authors are asked to incorporate the agreed revisions and the necessary additional justifications in the rebuttal/discussion to their final version.